# An effective spatial relational reasoning networks for visual question answering

**Xiang Shen**[1☯], **Dezhi Han**[1☯]*, **Chongqing Chen**[1‡], **Gaofeng Luo**[2‡], **Zhongdai Wu**[3‡]

**1** College of Information Engineering, Shanghai Maritime University, Pudong, Shanghai, China, **2** College of Information Engineering, Shaoyang University, Shaoyang, Hunan, China, **3** COSCO Shipping Technology Company Limited, Shanghai, China

☯ These authors contributed equally to this work.
‡ CC, GL and ZW also contributed equally to this work.
* dzhan@shmtu.edu.cn

**Data Availability Statement:** The data underlying the results presented in the study are available from (include the name of the third party https://visualqa.org/vqa_v2_teaser.html) The codes of our

## Abstract

Visual Question Answering (VQA) is a method of answering questions in natural language based on the content of images and has been widely concerned by researchers. The existing research on the visual question answering model mainly focuses on the point of view of attention mechanism and multi-modal fusion. It only pays attention to the visual semantic features of the image in the process of image modeling, ignoring the importance of modeling the spatial relationship of visual objects. We are aiming at the existing problems of the existing VQA model research. An effective spatial relationship reasoning network model is proposed, which can combine visual object semantic reasoning and spatial relationship reasoning at the same time to realize fine-grained multi-modal reasoning and fusion. A sparse attention encoder is designed to capture contextual information effectively in the semantic reasoning module. In the spatial relationship reasoning module, the graph neural network attention mechanism is used to model the spatial relationship of visual objects, which can correctly answer complex spatial relationship reasoning questions. Finally, a practical compact self-attention (CSA) mechanism is designed to reduce the redundancy of self-attention in linear transformation and the number of model parameters and effectively improve the model's overall performance. Quantitative and qualitative experiments are conducted on the benchmark datasets of VQA 2.0 and GQA. The experimental results demonstrate that the proposed method performs favorably against the state-of-the-art approaches. Our best single model has an overall accuracy of **71.18**% on the VQA 2.0 dataset and **57.59**% on the GQA dataset.

## 1. Introduction

Visual Question Answering (VQA) [1] is an emerging cross-field of Computer Vision (CV) and Natural Language Processing (NLP) in Artificial Intelligence (AI). It predicts the correctness of the question based on the image input by the user and the content of the image-related question answer, which is a very challenging task. It requires the model to be able to reason

models are available at https://github.com/shenxiang-vqa/SRRN.

**Funding:** This research is supported by the National Natural Science Foundation of China (Grant No. 61873160) https://www.nsfc.gov.cn/. This research is also supported by Scientific Research Fund of Hunan Provincial Education Department (Grant No. 21A0470) http://kxjsc.gov.hnedu.cn/. The funders had no role in study design, data collection and analysis, decision to publish, or preparation of the manuscript.

and understand images and text simultaneously. Their goal is to give machines the ability to understand vision and language like humans. In recent years, various multi-modal analysis tasks have emerged, breaking the boundaries between vision and language. Language and vision are widely used in computer vision tasks (*e.g.*, image captioning [2], visual descriptions [3], cross-modal information retrieval [4–6], visual question answering [1]), etc. Compared with other tasks, VQA is a method that requires the model to fully understand the input images and answer questions in natural language. At present, VQA tasks are also applied to real-life scenarios, such as assisting the blind and early childhood education, which has a wide range of practical applications. Considering the challenges and significance of VQA, visual question answering has received more and more research and attention in the intersection of computer vision and natural language processing.

In recent years, researchers have explored the multi-modal learning and visual reasoning of text and image features. The most advanced VQA method [7–11] is mainly focused on learning the multi-modal joint representation of images and questions. Specifically, the early proposed visual question answering model uses Convolutional Neural Network (CNN) to extract global features of an image and a Bag-of-words (BOW) model to extract text features of the question. After obtaining the global features of the image from the visual feature extractor, multi-modal fusion is used to learn a joint representation, which represents the alignment between each region and the question. Then input this joint representation into an answer prediction module to produce a correct answer. However, the use of global image features as the visual input of the model may introduce noisy information. In addition, the joint embedding learning method only maps the question and the image and lacks the process of reasoning, which leads to the lower accuracy of the model's answer. Researchers later introduced an attention mechanism to use image features and the distribution of questions to answers to alleviate noise information and make the model have simple reasoning capabilities. For example, Yang et al. [11] proposed a stacked attention network, which hierarchically focuses attention and locates the image region iterative. Lu et al. [8] proposed a hierarchical co-attention model, and learning the co-attention of vision and text simultaneously is more conducive to the fine-grained representation of images and questions to predict answers more accurately. Yu et al. [12] used the complementarity between visual attention and semantic attention to propose a novel multi-level attention network to enhance the fine-grained analysis of image understanding; Anderson et al. [13] proposed to detect salient objects in images for the first time and then use the top-down attention mechanism to learn object-level attention weights; Kim et al. [14] proposed a bilinear attention network and discussed high-order multi-modal fusion strategies to combine text information with visual information better; In addition, the researchers also proposed co-attention module BAN [14], DCN [15], DFAF [16], MCAN [17], which can stack these models to effectively capture the difference between the visual domain and the language domain. High-level information to obtain fine-grained multi-modal interaction features. The state-of-the-art performance has been tested on the benchmark dataset. However, these co-attention modules are still insufficient to model the complex inference features required for VQA tasks.

The VQA model introduced above mainly focuses on the attention mechanism and modal fusion strategy. The co-attention mechanism can only realize simple implicit relational reasoning. However, most VQA tasks require understanding and explicit reasoning on the input of the picture by the user. Visual-spatial relationship reasoning plays a significant role in modern VQA tasks. However, the model described above does not consider the spatial position relationship of visual objects but integrates the position features of the object into the visual features of the object, which leads to the lack of relationship in the model reasoning ability. It is very difficult to model the spatial relationship of visual objects. Because these targets are

Question：What is the animal in the picture?
Answer：giraffe

Question：What's under the cat?
Answer：laptop

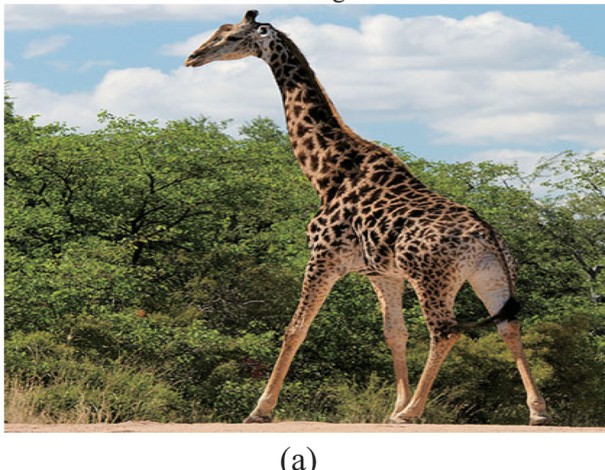

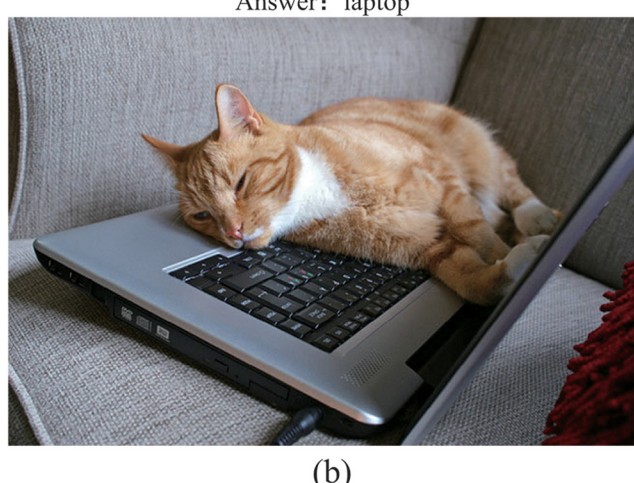

(a)

(b)

**Fig 1. Example of visual question answering tasks.**

located anywhere in the image, have different scales, belong to different categories, and different images have different numbers of visual objects. For example, responding to the question "What is the animal in the picture?" in Fig 1(a). The VQA model only needs to detect the "giraffe" object in the image and is not necessary to understand the entire image's content, as shown in Fig 1(b). "What is under the cat?" The model must first locate the two objects, cat and laptop, then model the spatial relationship between the objects, and fully understand the spatial concept of "under" before making the correct answer. We believe that combining the deep co-attention mechanism with the spatial relational reasoning mechanism can further improve the performance of the VQA model.

Inspired by related works [17–21], we designed an effective spatial relational reasoning network (SRRN) for visual question answering based on a deep co-attention mechanism. A novel sparse attention mechanism is introduced into the question encoder, which can explicitly select compelling question features and strengthen image features' selection in guided attention to image features. The sparse attention encoder can avoid the introduction of irrelevant information into the model and enhance the interaction between modalities. In addition, we also designed an efficient visual object spatial relationship reasoning network based on a graph neural network, which can capture the relationship between static objects and objects other than region detection. In this way, object spatial relationship modeling is realized so that the model can obtain fine-grained image features and visual object concepts, which can be used to answer complex spatial relationship reasoning questions. Considering the complexity and efficiency of model calculation, we propose a compact self-attention (CSM) mechanism based on the existing model, which effectively reduces the linear redundancy in the linear variation process, improves the model calculation efficiency and effectively improves the model precision and performance. The quantitative and qualitative experimental results on the VQA 2.0 [22] and GQA [23] datasets show that the SRRN model is an effective object spatial relationship reasoning network. In summary, our contributions are as follows:

1. A novel spatial relationship reasoning network model is proposed, effectively modeling visual objects' spatial position relationship and object attribute relationship. The SRRN

model can simultaneously model visual object semantic features and spatial position relationship features, which is vital for the model to predict the correct answer in VQA tasks.

2. A sparse attention encoder is designed to avoid the introduction of irrelevant information into the model when modeling question feature relationships while also improving the ability of modal interaction. A sparse attention encoder also dynamically captures each question's most relevant visual object features.

3. A compact self-attention (CSA) optimization algorithm reduces the linear redundancy in linear change. This method can effectively improve the computational efficiency of the model and improve the model's accuracy based on the initial model.

4. The SRRN model was tested on the benchmark datasets VQA 2.0 and GQA to obtain more satisfactory accuracy, and the ablation experiment analyzes the best performance of the SRRN model. Visual examples further reveal the interpretability of the model.

## 2. Related works

### 2.1 Visual question answering

The VQA task has received increasing attention from researchers in the past few years. The traditional VQA research method maps the image and question features into a common high-dimensional space and sends them to the classifier for classification tasks. The representation method of global features will introduce noise and make the model ineffective. Many researchers propose an attention mechanism in the VQA task to improve the model effect. For example, [7–10, 24, 25] use CNN-based networks to explore various image attention mechanisms to localize question-related regions. At the same time, there are also related works [8, 9, 26] that propose methods of using question-guided image attention and image-guided question attention and modal fusion of the extracted image and question features through the decoder and encoder. Multimodal fusion is crucial in VQA models. Traditional multimodal fusion methods map image features and question features to joint space embeddings. Recent studies [14, 27–30] have explored complex multimodal fusions way. With the advent of pre-trained models, multi-tasking in vision and language can effectively promote alignment between different modalities so that the models show good performance. In the previous pre-training process, the visual and language module was pre-trained independently in early visual question answering methods, failing to capture the connection between the visual dataset and the language dataset. In recent years, researchers have applied models based on Transformer structure to visual language tasks, and their representative works include ViLBERT [31], VLBERT [32], LXMERT [33], VisualBERT [34]. While these models achieve great results, they can significantly improve the model's performance by pre-training the base model and transferring it to downstream tasks based on large-scale visual and question datasets. However, in the research of network models for VQA tasks and downstream tasks, utilizing the end-to-end approach [7–9, 12, 16, 17, 20] to train network models can better capture the modal information of images and texts and can effectively improve model performance.

Explicit modeling of object interaction represented by graphs has attracted more and more attention. The object-relational reasoning ability of the VQA model is the core component of modern AI. At present, the main reasoning is divided into implicit reasoning and explicit reasoning. In [35], a graph-based method is proposed, which combines questions and abstract images with graph neural networks. It shows the great potential of graph neural networks in the VQA task. [36, 37] use simple graph neural networks to infer the object relationships between image regions implicitly. In recent years, explicit reasoning methods to model images

have also been widely used in many tasks [37–39]. Many researchers use graph neural networks to focus on counting problems in VQA tasks. [40] proposed an outer product computing graph network based on feature attention weights. This method only uses feature changes to improve the baseline model's performance counting ability. [41] proposed an iterative method based on object similarity to improve the counting ability and interpretability of the model. These two methods aim to eliminate duplicate target detection to evaluate the model based on counting. Wang et al. proposed VQA-Machine [42], which clarified the semantic relationship between objects. Although these models can significantly improve the performance of the VQA model, these models cannot simultaneously model the visual object space and the semantic representation of the visual object. The test results of the SRRN model on the VQA 2.0 and GQA datasets show that our model has a significant improvement in counting ability and can model the relationship between visual-spatial objects and the visual representation of objects simultaneously.

## 2.2 Attention mechanisms

The basic idea of the attention mechanism in the VQA model is to pay attention to specific visual regions and specific text in the image and provide more practical information for the vision to answer questions for the input questions related to the image. Bahdanau et al. [43] first used the attention mechanism to improve the neural machine translation (NMT) task. After that, the attention mechanism was also widely used in the VQA model and became an essential part of the VQA task. At present, the attention mechanism of the VQA task is mainly classified into three categories. The first category primarily focuses on the image region by questionable guidance. Most of this method uses top-down image attention and expresses question features and image features as vector elements to convey the concept of visual objects in the image region. The visual representation of the attention is generated by calculating the average value of all the visual features of interest. Yang et al. [11] proposed that Stacked Attention Networks (SANs) use the obtained context vectors to continuously pay attention to the regions in the image to obtain more accurate image features. Zhu et al. [44] combined CNN with Long Short-term Memory (LSTM) to generate an attention map for each output word. Researchers have proposed that DAN [24] uses one or more support and opposition paradigms of the differential attention network to obtain a different attention region, making it more like human attention. The second category is the cooperative attention mechanism. This attention mechanism considers visual attention and contextual attention (which words are more important in the question), using the hierarchical model of the co-attention model. The image representation can guide the attention of the question, and the performance of the question can guide the attention of the image. Nam et al. [9] proposed a dual attention network to collect necessary information through multi-step processing of specific areas in the image and keywords in the question. However, because these attention models learn multimodal coarse interaction examples, it is difficult to infer the correlation images and questions between them. Yu et al. [17] proposed the Deep Modular Co-Attention Networks (MCAN) model that overcomes the shortcomings of the model's dense attention (that is, the relationship between words in the text) and the relationship between regions in the image) in each mode at the same time. The third category is the target detection attention mechanism. Anderson et al. [13] used the target detection network Faster R-CNN to achieve bottom-up attention, segmented the image into specific objects for screening, and selected the first $K$ proposals in the picture as visual features. Lu et al. [45] proposed to combine free-form attention and detection attention to better expand the breadth of detection categories.

## 2.3 Visual relational reasoning

With deep learning and machine learning development, researchers are also exploring visual objects in the VQA task. Early works [46–48] proposed that the target relationship (*e.g.*, position and size [48]) was invoked as post-processing steps for target detection. The processing step is a method of re-scoring the detected target. In addition, some previous works [49, 50] also explored the spatial relationship between visual objects to help the model better understand the positional relationship of objects in the image. Recently, visual relational reasoning has been introduced to VQA tasks, which can help the model better answer images and questions that require logical understanding. It has received extensive attention from researchers. For example, visual object spatial relationship reasoning helps the cognitive task of image mapping to subtitles [51, 52] and improves image search [53, 54] and target localization. Recent visual object-relational reasoning [55, 56] focuses more on semantic relational reasoning rather than object-spatial relational reasoning. At present, some neural networks are also used for visual relationship prediction tasks [57, 58]. Most of the existing visual object-relational reasoning researches focus on implicit reasoning relations; they do not use explicit semantics or spatial relations to construct graphs. They model the interaction of objects by capturing implicitly on all attention modules or through fully connected graphs of high-level input images [19, 59]. For example, in [60, 61], the bilinear fusion method MuRel cell is introduced to model the object relationship. Yu et al. [61]designed a visual relationship reasoning relationship module to reason about paired and intra-group visual relationships between visual objects to enhance visual representation at the relationship level.

## 3. Methodology

The following is the question definition of the VQA task: Given a question *q* based on the picture *I*. The goal is to predict an answer $\hat{a} \in A$ that matches the ground-truth answer. As in the common literature in VQA, it is defined as a classification problem:

$$\hat{a} = \underset{a \in A}{arg} \ \max \ p_\theta(a|I, q) \tag{1}$$

where $p_\theta$ is the training model.

In this section, we will describe the detail of the SRRN model. The overall flowchart of the SRRN model is shown in Fig 2. It is mainly composed of question and image feature extraction, visual object semantic reasoning model and spatial relation reasoning module, and modal fusion and answer prediction module. We first describe how to extract the image and question features, then explain the visual object spatial relationship reasoning module and semantic reasoning module. Finally, the modal fusion and answer prediction module will be described. In the visual object semantic reasoning module, we will introduce the sparse attention mechanism encoder and the compact self-attention (CSA) optimization strategy in the co-attention module.

### 3.1 Question and image representation

The MCAN [17] model similarly makes all questions have the same length. We first trim each input question to a maximum of *S* words by simply discarding the extra words of the question longer than *S* words. Each word is transformed into a vector representation and pre-trained on a large-scale corpus to obtain a 300-D GloVe feature vector [62] into a word vector. Then the question is converted into a sequence of word embeddings $\{e_1, e_2, \cdots e_S\}$, which are then

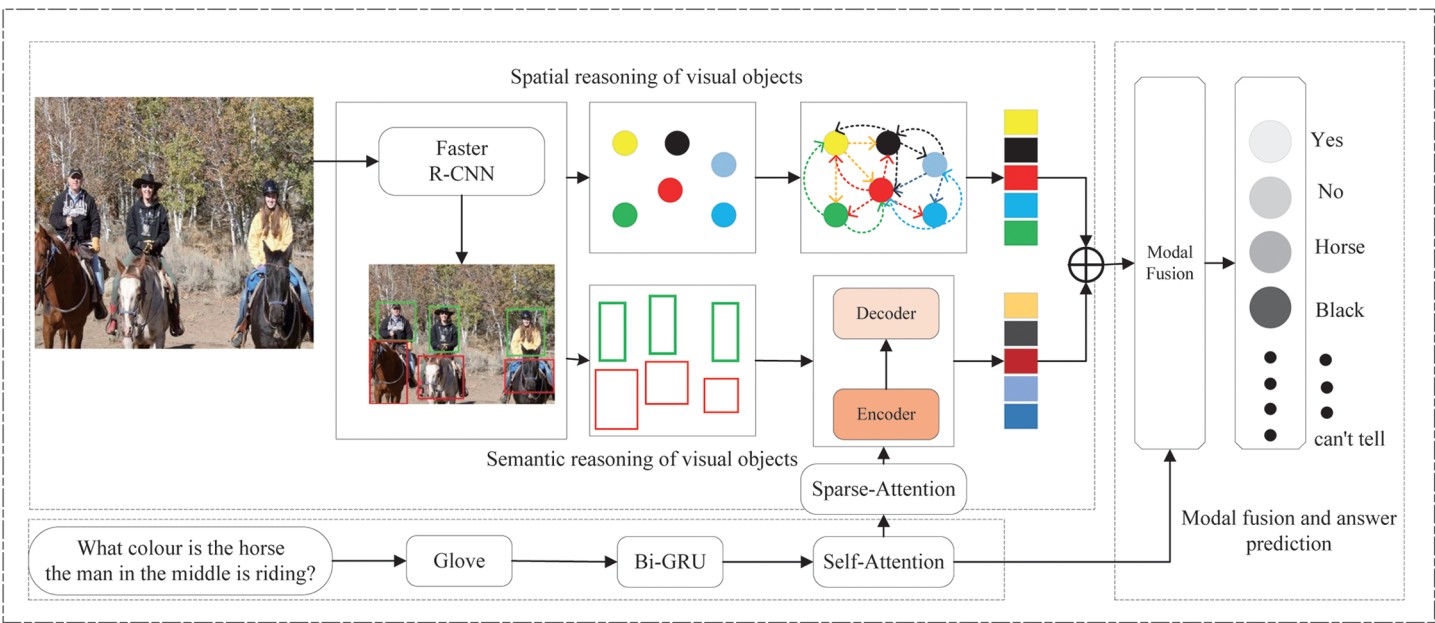

**Fig 2. Overall flowchart of the proposed SRRN model.**

passed through a bi-directional GRU (Bi-GRU) to output the word representation as follows:

$$\vec{q}_n = Bi - GRU(\vec{q}_{n-1}, e_n) \tag{2}$$

$$\overleftarrow{q}_n = Bi - GRU(\overleftarrow{q}_{n+1}, e_n) \tag{3}$$

where $\vec{q}_n$ is the output value of the forward hidden layer, and $\overleftarrow{q}_n$ is the output value of the backward hidden layer. Each question can be represented by a matrix $Q_q = \{q_1, q_2, \cdots q_S\} \in R^{d_q \times S}$, where $q_n = [\vec{q}_n, \overleftarrow{q}_n]$, and $[\cdot, \cdot]$ denotes concatenation.

Inspired by bottom-up attention [14], we use the pre-trained Faster R-CNN ResNet-101 [63] network to extract the target detection frame on the input image, and at the same time it is used to identify a sequence of visual objects $V_I = \{v_i\}_{i=1}^{K}$. The defined visual object is related to the visual representation vector $v_i \in R^{d_v}$ and the boundary feature vector $b_i \in R^{d_b}$ at the same time. Set ($K = 36$, $d_v = 2048$, $and\ d_g = 4$) in the experiment. Each $g_i = [x, y, w, h]$ corresponds to a 4-dimensional space coordinate, where ($x, y$) represents the coordinate point of the upper left corner of the bounding box, and ($h, w$)corresponds to the height and width of the box. $K$ salient targets are obtained by selecting candidate boxes, and the $i$-th visual object feature is denoted as $v_i \in R^{d_v}$. The image output feature is $V_I \in R^{K \times 2048}$, and the setting is the total number of target detection features from some comparative experiments and computing resource conditions. Considering the better performance and higher computational efficiency, set $K = 36$. In the experiment, we use linear change $V_I$ to make the image feature dimension consistent with the question feature dimension. Therefore, the final image feature is $V_I \in R^{K \times 512}$.

## 3.2 Spatial reasoning of visual objects

The visual object spatial relationship reasoning proposed in this paper can dynamically capture the relationship between objects in an image. For VQA tasks, different question types have

different visual object relationships. The spatial relationship features of visual objects are mainly composed of appearance features and geometric features. The visual object representation is mainly the focused image feature output by the co-attention module, and the geometric feature is the 4-dimensional visual object bounding box represented by $g_i$. Given that there are $K$ visual objects for self-attention learning, the generated hidden layer relationship feature $\{v_i\}_{i=1}^{K}$ is used to represent the relationship between the target object and adjacent objects. The graph reasoning attention mechanism formula of each object spatial relationship is as follows:

$$v_i = \sigma\left(\sum_{j\in N_i}\alpha_{ij}.Wv'_j\right) \qquad (4)$$

For different VQA tasks, the definition of the attention coefficient $\alpha_{ij}$ in formula (4) is also different, where the projection matrix $W \in R^{d_h \times (d_q + d_v)}$ and the target domain object are different. $\sigma(\cdot)$ is the activation function.

Since the construction of the reasoning graph between visual objects is a fully connected graph, all $N_i$ includes the object itself and all the visual objects in the image. Inspired by [19], we designed the attention weight $\alpha_{ij}$ to rely on the visual weight $\alpha_{ij}^v$ and the object bounding box $\alpha_{ij}^g$. The specific equation is as follows:

$$\alpha_{ij} = \frac{\alpha_{ij}^g \cdot \exp(\alpha_{ij}^v)}{\sum\limits_{j=1}^{K}\alpha_{ij}^g \cdot \exp(\alpha_{ij}^v)} \qquad (5)$$

where $\alpha_{ij}^v$ represents the similarity between the object and the object's position, and the calculation of $\alpha_{ij}^v$ uses the scaled dot product [64]. The specific equation is as follows:

$$\alpha_{ij}^v = \left(Uv'_i\right)^T \cdot Vv'_j \qquad (6)$$

where $U, V \in R^{d_h \times (d_q + d_v)}$ is a projection matrix, $\alpha_{ij}^g$ is to calculate the relative geometric position between the object and the object, the specific equation is as follows:

$$\alpha_{ij}^g = \max\{0, w \cdot f_g(g_i, g_j)\} \qquad (7)$$

where $f_g(\cdot, \cdot)$ first calculates a 4-dimensional relative geometric feature $\left(\log\left(\frac{x_i - x_j}{w_i}\right), \log\left(\frac{y_i - y_j}{h_i}\right),\right.$ $\left.\log\left(\frac{w_j}{w_i}\right), \log\left(\frac{h_j}{h_i}\right)\right)$, and then embed it into dimensional feature by calculating the cosine function and sine function of different wavelengths. $w \in R^{d_h}$ convert the $d_h$-dimensional features into scalar weights. The model crops the scalar weight at 0 to limit the specific geometric relationship between the visual objects. $g_i$, $g_j$ are 4-dimensional space coordinates, which represent the relative geometric position relationship of visual objects.

In addition, to strengthen the spatial reasoning relationship features of visual objects, we have also extended the above graph attention mechanism and adopted multi-head attention. Using $M$ independent multi-head attention mechanisms and connecting their output features, the following features are obtained:

$$v_i^* = \|_{m=1}^{M} \sigma\left(\sum_{j\in N_i}\alpha_{ij}^m \cdot W^m v'_j\right) \qquad (8)$$

Finally, $v_i^*$ is added to the original visual features $v_i$, as the final reasoning feature of spatial object relation.

## 3.3 Semantic reasoning of visual objects

The visual semantic reasoning is formed by stacking encoders and decoders similar to the MCAN module. Unlike MACN, we use a sparse attention mechanism in the encoder, which helps to capture text context information better. At the same time, the use of a sparse attention encoder can prevent irrelevant information from being introduced into the model, which helps to enhance the robustness of the model and the semantic reasoning ability of visual feature objects. Experiments verify that the sparse attention encoder is effective in retaining important features and removing noise.

**3.3.1 Sparse mechanism encoder.** We first introduce the sparse attention mechanism encoder, as shown in Fig 3. The sparse attention mechanism encoder is a simplified transformer model, including a multi-head dot-product attention layer [64] and several fully connected layers. The question of traditional transformer encoder self-attention can establish a long-term dependence model. However, when modeling question features for self-attention learning, context-irrelevant information is also introduced into the model to distract the attention weight of the model. In order to solve this problem, we use a sparse attention mechanism encoder in the question encoder, which can improve the concentration of attention on the global context by displaying and selecting the most relevant segments. It helps the question features important guide regions of the image.

Fig 3 shows the structure of a sparse attention mechanism encoder. The question features get different vector values, key values and query values through linear transformation, in which the similarity of query values and key values determines the weight of similarity of question words. For the convenience of calculation, we separately set the input of sparse dot

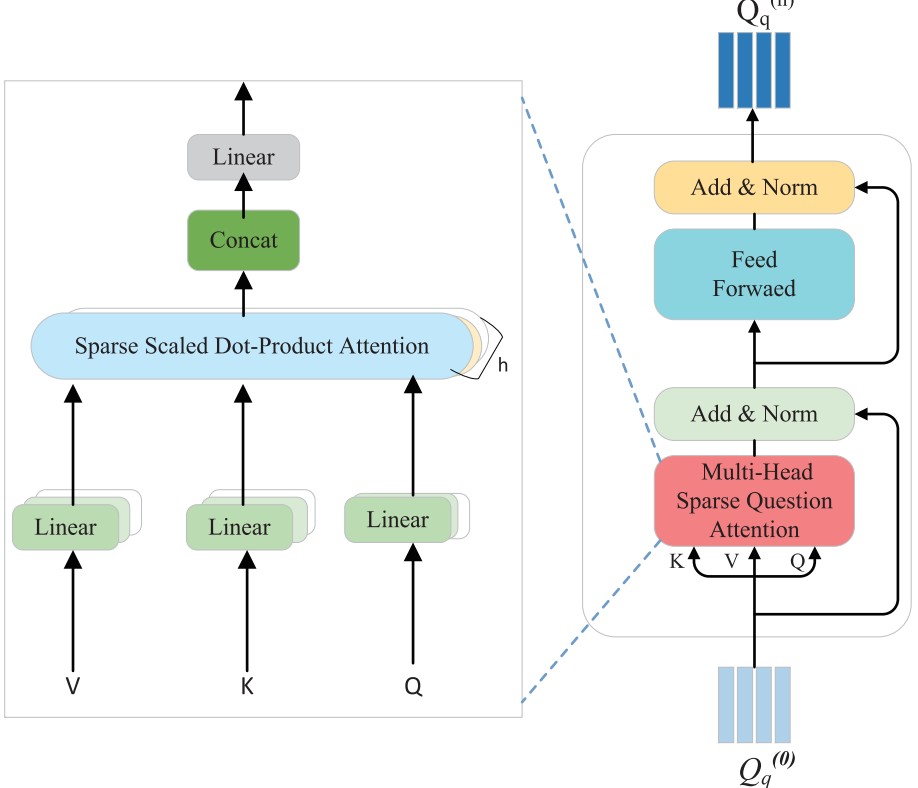

**Fig 3. Sparse attention mechanism encoder.**

product attention including query $Q_E[l_Q, d]$, key $K_E[l_K, d]$, and value $V_E[l_V, d]$. Firstly, the scaling dot-product formula is used to calculate the similarity between the query value and the key value to obtain a high-dimensional matrix, and then divide by get an attention matrix score, the specific equation is as follows:

$$W = s(Q_E, K_E) = \frac{Q_E K_E^T}{\sqrt{d}} \tag{9}$$

We assume that the higher the score calculated in the $W$ matrix, the higher the correlation between word features. In practice, for simultaneous estimation $c$ query value function, which we embed into a matrix $Q_E \in R^{c \times d}$; Similarly, $t$ key-value functions are embedded in the matrix $K_E \in R^{t \times d}$. Obtaining $W \in R^{c \times t}$ is a weight matrix as shown in Fig 4.

The principle of the sparse attention mechanism is to eliminate the words with low weights learned by the initial scaled dot-product attention model self-attention, which is used to guide essential image regions related to the question. We assume that the higher the score, the higher the correlation, and the sparse attention masking operation $M_{ij}$ is performed on $W$ to select the most critical $\delta$ contribution elements. Specifically, we select the most significant $\delta$ element and record their position in the position matrix $(i, j)$, where $\delta$ is a parameter. $\delta$-th is the row with the most considerable value in the $i$-th item $a_i$. If the value of the $j$-th component is greater than $a_i$, the position $(i, j)$ is recorded. We connect the thresholds of each row to form a

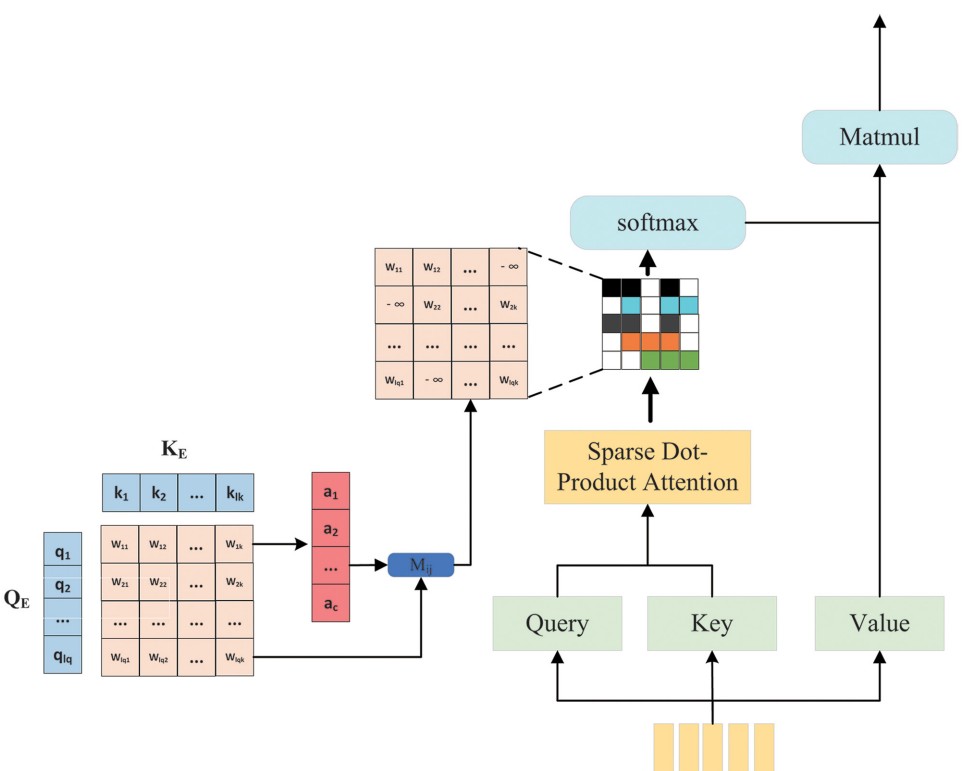

**Fig 4. The sparse attention mechanism.**

vector $A = [a_1, a_2, \ldots, a_c]$. The mask function $M_{ij}$ is defined as follows:

$$
M_{ij} = \begin{cases} W_{ij}, & if \ \ W_{ij} \geq a_i \\ -\infty, & if \ \ W_{ij} < a_i, \end{cases} \tag{10}
$$

$$
\bar{M} = softmax(M_{ij}) \tag{11}
$$

where $\bar{M}$ refers to the standardized score, since scores smaller than the previous maximum score are assigned $-\infty$ by the masking function $M_{ij}$, their normalized scores, the probability is approximately 0. The output representation of self-attention after selection can be calculated as:

$$
F = \bar{M}V \tag{12}
$$

$F$ is the expected value of sparse distribution, and the sparse attention mechanism can get more concentrated attention. This attention mechanism can be extended to context attention, similar to the common self-attention mechanism but differs in that $Q_E$ is not a linear change of the original context but a decoding state. In order to further improve the semantic representation ability of visual objects, inspired by multi-head attention [60], each head uses a sparse dot product attention function to calculate the weight of the question word in the input encoder. Unlike the ordinary multi-head attention mechanism, we directly choose the feature weights of important issues that are directly discarded if lower than the threshold. Finally, we can get the most critical weight information of the question. The calculation equation is defined as follows:

$$
MHSAtt = (Q_E, K_E, V_E) = Concat(head_1, \cdots head_h)W^o \tag{13}
$$

$$
head_i = SAtt(Q_E W_i^Q, K_E W_i^K, V_E W_i^V, \delta) \tag{14}
$$

where $W_i^Q$, $W_i^K$ and $W_i^V \in R^{d \times d_h}$ are the projection matrices of the $i$-th head, and $W^O \in R^{h \times d_h \times d}$ is the learned weight matrix. SAtt(.) represents dot-product self-attention using a sparse attention mechanism.

**3.3.2 Co-attention modular.** The co-attention module in the SRRN model is based on the scaled dot product attention, and the scaled dot product attention is a mapping. It is embedded in three matrices, representing the embedded query, key, and value vectors, and these matrices are named $Q, K, V \in R^{L \times d}$ by convention. Where $L$ is the sequence length of the input tag, and $d$ is the hidden dimension. Then the proportional dot product attention can be defined as:

$$
Att(Q, K, V) = soft \max\left(\frac{QK^T}{\sqrt{d}}\right)V \tag{15}
$$

Inspired by the work [21], we carefully examined and reviewed linear transformations. Since $Q$ and $K$ vary linearly from the same input, the weight matrices $W_K$ and $W_Q$ are entangled in the gradient backpropagation, a basic redundancy in the conventional self-attention mechanism. To avoid this redundancy, we propose to contribute the weights of $K$ and $V$ to achieve weight redundancy in the process of self-attention. To avoid this redundancy, we propose to contribute the weights of $K$ and $V$ to achieve weight redundancy in the process of self-attention. We set $W_k = W_v$ and found through experiments that the compact self-attention

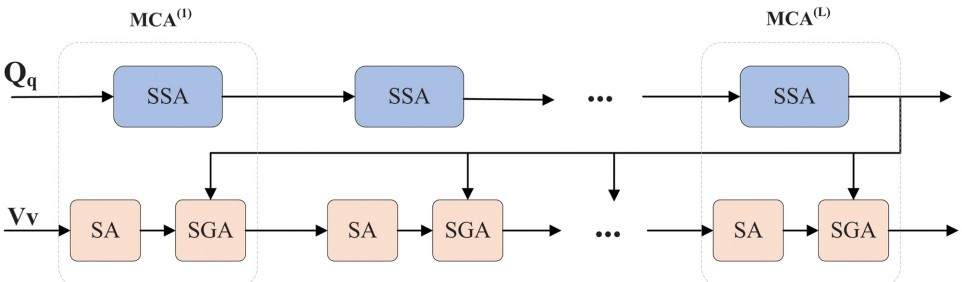

**Fig 5. Co-attention structure diagram.**

mechanism has a good optimization effect. The specific change equation is as follows:

$$Att(Q, K, K) = soft\max\left(\frac{QK^T}{\sqrt{d}}\right)K \tag{16}$$

Fig 5 shows the structure of the multi-modal co-attention module, which is mainly composed of a stack of encoders and decoders. The encoder uses a sparse self-attention (SSA) mechanism to capture important question features to guide attention to important question regions features in the image. $Q_q$ is input into the SSA unit as a question feature, and essential question features are learned through self-attention. Specifically, The SSA unit consists of two sub-layers (see Fig 4). $Q_q = \{q_1, q_2, \cdots, q_s\} \in R^{d_q \times S}$ is used as the question feature input to calculate the relationship between each word $<q_i, q_j>$, and then the sparse question feature is input into the fully connected layer to obtain the weight between each word. The sparse encoder question feature output matrix $F_1$ can be expressed as:

$$F_1 = MHSAtt(Q_q, K_q, K_q) \tag{17}$$

$$= Concat(head_1, \cdots, head_h)W^O \tag{18}$$

$$head_i = SAtt(Q_q W_i^{Q_q}, K_q W_i^{K_q}, K_q W_i^{Kq}, \delta) \tag{19}$$

where $Q_q W_i^{Q_q}$ and $K_q W_i^{K_q}$ are the sparse question feature matrix of the $i$-th head. is the question feature selecting a parameter, and the feedforward layer further transforms the question features. The final features are as follows:

$$FFN(F_1) = \max(0, F_1 W_1 + b_1)W_2 + b_2 \tag{20}$$

where $W_i$ and $b_i$ represents weight coefficient and biased variable respectively.

The decoder has three sub-layers, mainly composed of SA unit and SGA unit. Specifically, SGA units in the first layer need sparse question features. $Q_q = \{q_1, q_2, \cdots, q_s\} \in R^{d_q \times S}$ and $V_v = \{v_1, v_2, \cdots, v_k\} \in R^{k \times 2048}$ as an input, sparse question features are used to guide the features of the concerned image. The second layer is self-attention learning of image and question features and output to the third layer full connection layer. Final output question and image features $<q_i, v_j>$ in the experiment, we tried to add a sparse image self-attention unit before the first layer, but the experimental results were not satisfactory. In addition, we also use linear transformation to keep the dimension of question features consistent with that of image

features. The specific equation is as follows:

$$F_2 = MHSAtt(Q_q, K_v, K_v) \tag{21}$$

$$= Concat(head_1, \cdots, head_h)W^O \tag{22}$$

$$head_i = Att(Q_q W_i^{Q_q}, K_v W_i^{K_v}, K_v W_i^{K_v}) \tag{23}$$

Feed forward the questions and image features $F_2$:

$$FFN(F_2) = \max(0, F_2 W_1 + b_1)W_2 + b_2 \tag{24}$$

where $W_1 \in R^{512 \times 2048}$, $W_2 \in R^{2048 \times 512}$, and $b_1, b_2 \in R^{512 \times 2048}$ are projection matrixes.

Take the image mentioned above feature $V_v$ and question feature $Q_q$ as input. Through deep cascading $L$-layer $MCA$ (represented as $MAC^{(1)}, \ldots, MAC^{(L)}$), a deep co-attention model is formed to transfer input features and perform deep co-learning. The input features of $MAC^{(L-1)}$ are $Q_q^{(l-1)}$ and $V_v^{(l-1)}$. These features are further passed to the $MAC^{(L+1)}$ layer as input in a recursive manner. The specific equation is as follows:

$$[Q_q^L, V_v^L] = MCA^L([Q_q^{(L-1)}, V_v^{(L-1)}]) \tag{25}$$

We set the input features of $MAC^{(1)}$ as $Q_q^{(0)} = Q_q$ and $V_v^{(0)} = V_v$ respectively.

## 3.4 Modal fusion and answer predictions

The question feature $Q_q$ and the image feature $V_v$ output by the encoder and decoder. After the sparse attention mechanism encoder, co-attention learning, the question feature, and the image feature contain the most significant feature value, which provides the weight information of the word and image region. Similarly, the spatial relationship feature of visual objects obtained by graph reasoning is denoted as $v_r^*$. Then we design an attentional reduction model with a two-layer $MLP(FC(512) - Relu - Dropout(0.1) - FC(1))$ to obtain its attended question feature $\bar{Q}_q$ and image feature $\bar{V}_v$. Specifically, we input the image features into the $MLP$ through the softmax function to calculate the attention weight value, and then multiply and sum each region image feature to get the final image feature. The equation is as follows:

$$\lambda = soft \max(MLP(V_v^L + V_r^*)) \tag{26}$$

$$\bar{V}_v = \sum_{j=1}^{m} \lambda_j (V_j^{(L)} + V_r^*) \tag{27}$$

where $\lambda = [\lambda_1, \lambda_2, \cdots, \lambda n] \in R^n$ are the learned attention weights. $L$ represents the number of stacked layers of MAC. Use the softmax function to obtain the attention weights related to the image and the question and normalize these weights in all regions. Finally, the image and question features from all regions are weighted through these attention weights, and the final weighted sum is used as the final visual feature and question feature. Use the linear multi-modal function to fuse the final question feature $\bar{Q}_q$ and image feature $\bar{V}_v$, and the fused feature is expressed as Eq 28:

$$f = layerNorm(W_v^T \bar{V}_v + W_q^T \bar{Q}_q) \tag{28}$$

where $f$ represents the fusion feature of the question and the image, and $W_v$ and $W_q$ are linear

projection matrices. Then the $f$ is passed through the non-linear activation function *Relu*, and the sigmoid function is used to classify the answer. In the training process, we use the binary cross-entropy (BCE) function as the loss function.

$$s = sigmoid(W_0 relu(W_f f)) \tag{29}$$

where $s$ represents the score of the candidate answers, and $W_f$ is the linear projection matrix. The candidate answers with the highest probability are selected as the prediction result. Finally, we use BCE as the loss function to train $N$ answer classifications.

$$N = \sum_{i}^{N} \gamma_i \log(s_i) + (1 - \gamma_i)\log(1 - s_i) \tag{30}$$

where $N$ is the size of the candidate set, and $s_i$ is the score predicted by the model for each candidate answer, $\gamma_i$ is the soft score that provides the answer in the dataset.

## 4. Experiments

All experiments in this paper are based on Linux Ubuntu 18.04 system, GPU is NVIDIA TITAN V 12GB, deep learning framework is Pytorch, and CUDA version is 10.0. This section first describes the VQA 2.0 dataset [22] and the newly introduced GQA [23] dataset in Section 4.1 to evaluate our proposed model. In Section 4.2, we describe the experimental setup details. Section 4.3 discusses the ablation experiment and displays the experimental results and experimental setting parameters. Section 4.4 respectively describes the results of our proposed SRRN model compared with the state-of-the-art results on the two datasets. Finally, we use several successful examples and failure examples to explain the model reasonably visually.

### 4.1 Dataset

Unlike pre-trained model datasets, models trained end-to-end utilizing VQA-specific datasets are more likely to capture and extract image and text features, which are helpful for the classification of downstream tasks of the model. However, the datasets of the pre-trained models come from various corpora, and the features learned from different corpora have generalization and generality. Using pre-trained datasets can also effectively avoid biases and language priors that exist in the dataset from interfering with model performance. For a fair comparison of the experimental results, this paper employs the VQA 2.0 and GQA datasets to train the model.

**VQA 2.0**: The SRRN model is training, validating, and testing on the VQA 2.0 [22] dataset, which is based on Microsoft COCO image data and is currently the most commonly used large-scale dataset for evaluating the performance of visual question answering models. It tries to minimize the effectiveness of the model learning dataset bias by balancing the answers to each question. The VQA 2.0 dataset contains 1.1M questions posed by humans. It consists of three parts: training set, validation set, and test set. Each valid piece of data is represented by a three-element question and answer group composed of the dataset (image, question, answer). The training set contains 82,783 images and 443,757 question and answer groups corresponding to the images. The verification set contains 40,504 images and a corresponding 214,354 question and answer groups, and the test set contains 81,434 images and 447,793 question and answer groups. According to the categories of answers, questions can be divided into three types: yes/no (Yes/No), count (Number) and Other. We show the results on *test-dev* and *test-standard* on the VQA evaluation server.

**GQA**: It consists of 22M questions generated from 113K images. Compared with VQA 2.0, more questions in the GQA data set require multi-step reasoning to balance the answers. About 94% of the questions require multi-step reasoning, and 51% need to query the relationship between objects. In addition to the standard accuracy measures, the authors of GQA have designed several new measures, including consistency, credibility, validity, and distribution. The higher the score, the consistency, effectiveness, and credibility in these indicators, but the lower score is conducive to promotion. The model is trained based on a balanced training split and a balanced verification split, and then the test split is tested on the evaluation server.

## 4.2 Details of the experimental setup

We implement our model with the Pytorch library on a machine with 4 Nvidia TITAN V 12GB GPUs. We set the dimension of the hidden layer in the proportional dot product attention to $d = 512$. The number of heads in the multi-head attention h is 8, and the number of dimensions of each head's output feature is $d/h = 64$. According to the suggestion in [28], the number of layers $L$ of the decoder and encoder is set to 6, and the structure of the feedforward layer is $FC(4d) - ReLU - dropout(0.1) - FC(d)$. The structure of the multilayer perceptron used to calculate the features of interest is $FC(d) - ReLU - dropout(0.1) - FC(1)$, where $ReLU$ is the activation function, and dropout is used to prevent overfitting. The number of visual reasoning features is set to $N_i = 16$. The dimension of the fusion feature $f$ is 1024. We use AdamW [65] ($\beta_1 = 0.9$, $\beta_2 = 0.999$) to train the SSRN model, set its batch size to 64 and use BCE as the loss function. The warm-up learning rate is $\min(2.5te^{-5}, 1e^{-4})$, where $t$ is the current epoch number starting from 1. The code implementation of all models proposed in this paper are based on PyTorch. In order to prevent the gradient from exploding, a gradient clipping strategy with a threshold of 0.25 is used; In order to stabilize the output and prevent over-fitting, each linear map is subjected to weight normalization and dropout processing.

## 4.3 Ablation study

This section mainly discusses choosing the optimal parameters and proving the validity and interpretability of the model. We designed different SRRN variant models and used *train+val +vg* for training on the VQA 2.0 dataset and tested them on the test dataset to obtain the results. Visual genome (*vg*) is a dataset, a knowledge base, an ongoing effort to connect structured image concepts to language. Moreover, we give a detailed discussion. Firstly, we explore the effects of using visual-spatial object reasoning, sparse encoder, and compact self-attention mechanism in the model in Section 4.3.1. Set different parameters in Section 4.3.2 to prove an effective sparse attention encoder.

**4.3.1 SRRN variants.** As shown in Table 1, we show the performance of different variants of the SRRN model. For the convenience of writing, denoted "SRRN-r" as the object spatial relation reasoning module in Table 1. The input and output dimensions of the object spatial relationship reasoning model of the SRRN model are the same. In the experiment, we use the stacked network module to explore the performance of the model. Among them, "SRRN-r1" and "SRRN-r2" stand for stacking one and two layers of object spatial reasoning modules, respectively. The experimental results found that although the effect of stacking two layers is better than that of stacking one layer, the calculation efficiency of the model will decrease due to the increase in the number of stacked layers, and the model network training parameters will also be increased. In order to reduce the number of parameters of the model and improve the calculation efficiency, the method of stacking one layer is used in the subsequent experiments. "SRRN-s" and "SRRN-c" indicate that the model uses a sparse encoder and a compact

**Table 1. Performance of different SRRN variant models on VQA 2.0 dataset.**

| Model | Yes/No | Number | Other | All |
|---|---|---|---|---|
| MCAN [17] | 86.82 | 53.26 | 60.72 | 70.63 |
| SRRN-s | 87.05 | 53.21 | 60.72 | 70.71 |
| SRRN-c | **87.14** | 53.56 | **61.21** | **71.02** |
| SRRN-r1 | 86.86 | 54.17 | 60.43 | 70.71 |
| SRRN-r2 | 86.78 | 54.57 | 60.57 | 70.69 |
| SRRN-r1-s | 86.79 | 54.20 | 60.51 | 70.78 |
| SRRN-r2-s | 86.84 | 54.82 | 60.54 | 70.73 |
| SRRN-r1-s-c | 86.97 | **55.02** | 60.80 | 70.92 |

self-attention mechanism, respectively. The experiments in Table 1 show that using the sparse attention mechanism encoder on the reasoning module will significantly improve the model effect. Because the sparse attention encoder can better capture the contextual information of the question, focus on the important image region when the question guides the focus on the image better to achieve the semantic reasoning of the visual object. The model we propose must process both the semantic features of the visual object and the spatial relationship reasoning feature to optimize the model. We optimize the model based on the sparse attention encoder and reasoning module and propose an effective compact self-attention optimization method. According to the experimental results, it can be seen that the compact self-attention optimization method is very effective. "SRRN-c" represents that the compact self-attention optimization method is directly used based on the MCAN model. Except for the accuracy of the "Number" type, there is no noticeable improvement. "Other" type indicators have excellent results. Through experimental comparison, it can be observed that our model reached 55.02% on the "Number" type, indicating that the use of the object spatial relationship reasoning module has excellent performance on the "Number" type. We adopted "SRRN-r-s-c" model finally. The comparison shows that our model combines visual semantic and spatial object relationship features with excellent VQA task performance. It is also a simple and effective visual question answering reasoning model.

We also discuss the accuracy of model training and the convergence speed of model training on the GAQ and VQA datasets. As shown in Fig 6, Fig 6(a) shows that the "SRRN-r-s-c" model is training on the GQA dataset using *train + vg*, and the accuracy of each epoch is verified on the *val* validation set. We found that training with the GQA dataset achieves the best results in the 9th epoch. As the training continues, the accuracy gradually decreases and tends to balance. The accuracy we compare in Table 4 uses the 9th epoch of training data. Fig 6(b) indicates the changes in the loss function of the "SRRN-r-s-c" model trained on the VQA 2.0 and the GQA datasets. Through experiments, we found that the model converges faster when using the GQA data laid down for training, and the VQA 2.0 dataset converges more slowly than the GQA dataset. GQA is a dataset for most VQA reasoning tasks, which illustrates our model's excellent spatial relational reasoning networks performance.

**4.3.2 SRRN parameter ablation.** In this section, we mainly discuss the selection of parameters used in the sparse attention encoder. We first discuss using the sparse attention encoder to select the appropriate hyperparameter. As shown in Fig 7, we verified the influence of different parameters on the model's performance. In the process of exploring the ablation experiment, we used the VQA 2.0 dataset *train + val + vg* to experiment based on the "SRRN-r-s-c" model in Table 1. From Fig 7, we find that when $\delta = 3$, the accuracy of "Other" and "All" is the best, and when $\delta = 8$, Number's accuracy is the best. The model's overall performance is

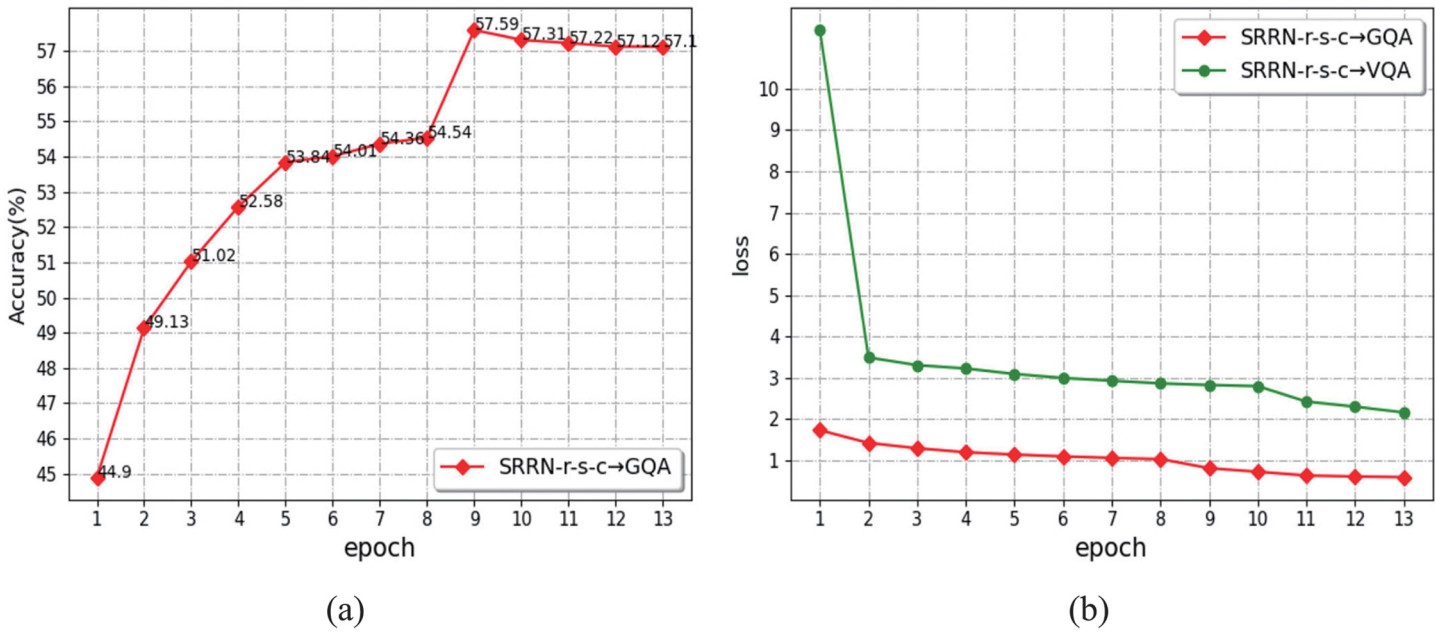

**Fig 6. The change of accuracy and loss function in the process of model training.**

better by combining the four indicators and setting the hyperparameter of sparse attention to $\delta = 8$.

Since the object semantic reasoning module of the SRRN model is based on the Transformer encoder and decoder stacked $L$ layers, we use different layers to verify the performance of the model. The experimental data is shown in Table 2. When verifying the number of layers, we use the *train* dataset for training and the *val* dataset for verification. Considering the computational time efficiency and cost, we set $L = 6$ for SRRN model experiment training.

**4.3.3 Number of model parameters comparison.** The parameters and accuracies of the SRRN model and the VQA pre-trained model are compared in Table 3. The SRRN model uses an end-to-end method to train on the VQA 2.0 dataset and obtains good results. Experiments show that the spatial object relation reasoning module can effectively improve the model effect. The amount and complexity of model parameters has always been a major concern for VQA tasks. Most of the existing VQA pre-training models can be fine-tuned on the VQA 2.0 dataset to achieve good results. In order to more effectively illustrate the performance and complexity of the SRRN model, we compare the SRRN model with the classical pre-trained model. Unified VLP [66], VilBERT [31], VisualBERT [34], VLBERT [32] are also pre-trained models using encoder and decoder. DFAF-BERT [16] and MLI-BERT [67] are based on end-to-end models using BERT as a pre-trained model. As shown in Table 3, the parameters of the "SRRN-r1-s-c" and "SRRN-r2-s" models are 58.19M and 58.98M, respectively, however most VQA pre-training models have larger parameters than the SRRN model. Besides, the SRRN model only needs 1 TITAN GUPs to complete the training.

## 4.4 Comparison with state-of-the-arts

In Table 4, we compare our model SRRN with the state-of-art model in VQA 2.0 dataset. A single model obtains all results. Table 4 is divided into three blocks in the row. In the first part, several feature models without Faster-RCNN are summarized. In the second part, the pre-

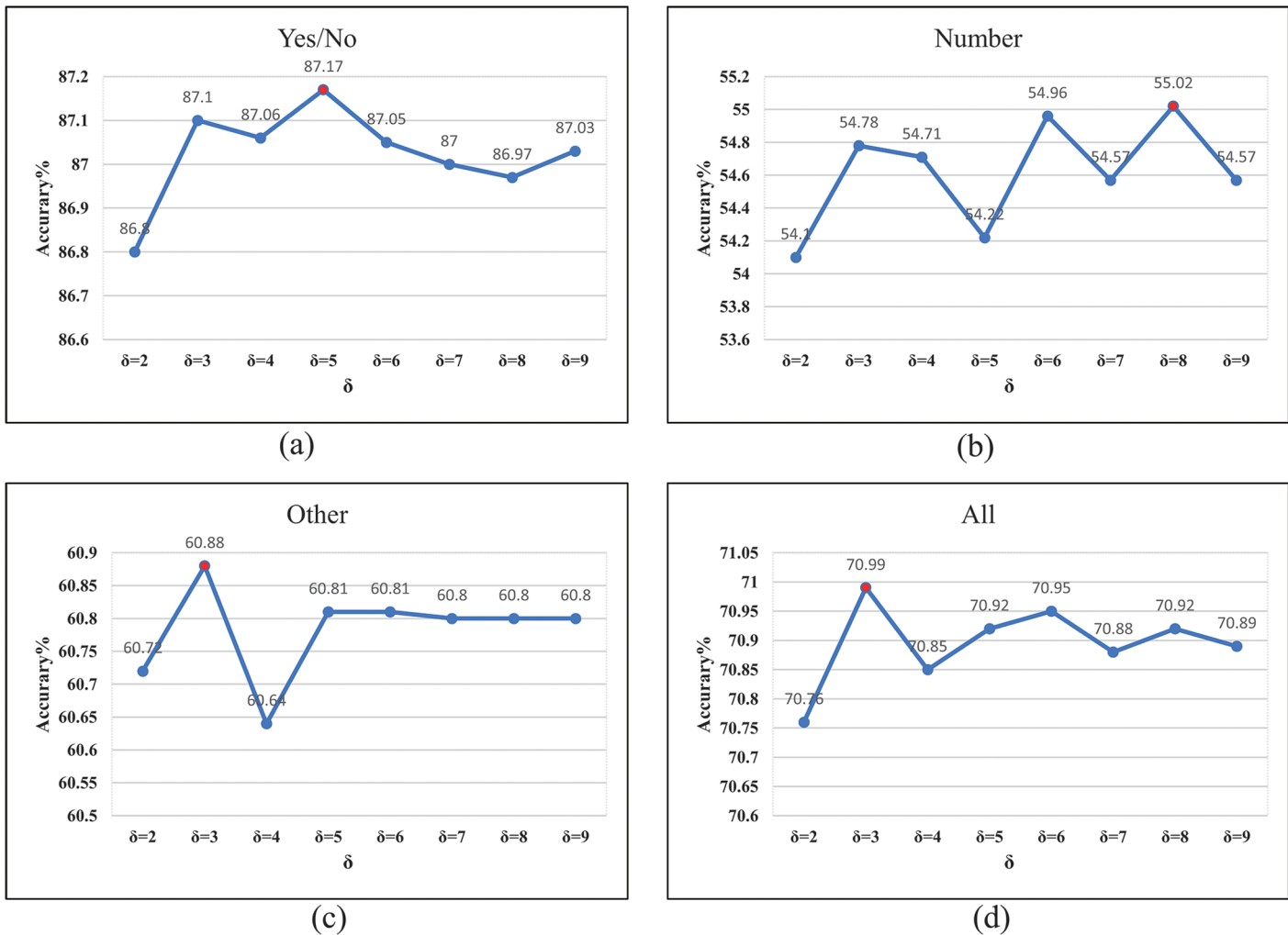

**Fig 7.** (a) The accuracy of "Yes/No" based on different parameters. (b) Accuracy of "Number" based on different parameter. (c) Accuracy of "Other" based on different parameters. (d) Accuracy of "All" based on different parameters.

trained Faster-RCNN is used to detect prominent targets and Glove is used to encode word vectors. Our results are in the last block, using the same Faster-RCNN and Glove for pre-training. Most model indicators are better than the existing advanced methods, especially "Number" counting, highlighting the importance of SRRN reasoning in VQA tasks.

Among these state-of-the-art models, Bottom-up [10] and Bottom-up+MFH [17] combine regional visual features with question-guided visual attention, which considers the biological

**Table 2. The performance of different layers of encoder and decoder.**

| L | Yes/No | Number | Other | All |
|---|--------|--------|-------|-----|
| 2 | 83.88 | 49.86 | 57.60 | 66.46 |
| 4 | 84.68 | 50.54 | 58.11 | **67.10** |
| 6 | **84.72** | 50.34 | **57.93** | 67.02 |
| 8 | 84.52 | **50.84** | 57.85 | 66.95 |

**Table 3. Comparison of pre-trained model parameters and SRRN model on the VQA 2.0 dataset.**

| Model | Parameters | Test-dev | Test-std |
|---|---|---|---|
| Unified VLP [66] | - | 70.50 | 70.70 |
| ViLBERT [31] | 218.9M | 70.55 | 70.92 |
| ViusalBERT [34] | 85.05M | 70.80 | 70.92 |
| VLBERT [32] | 134.8M | 71.16 | - |
| DFAF-BERT [16] | 173.2M | 70.59 | 70.81 |
| MLI-BERT [67] | 120.0M | 71.19 | 71.27 |
| SRRN-r2-s | **58.98M** | 70.73 | - |
| SRRN-r1-s-c | **58.19M** | 70.92 | 71.18 |

basis of attention. BAN [14] is a bilinear attention network that considers the bilinear interaction between the input multi-modality to use the question features and image features information fully. BAN-Counter [14] combines BAN with Counter [14]. The latter is a neural network structure that allows robust counting between object suggestions and further improves the accuracy of the model in counting indicators. MuRel [17] and ReGAT [20] used graphs to construct deep reasoning networks and graph reasoning networks based on the relationships between objects, achieving impressive results. Both ViLBERT [31] and VisualBERT [34] utilize the BERT architecture to extend a multimodal dual-stream task. The ViLBERT model is pre-trained on an automatically collected large-scale conceptual caption dataset by two proxy tasks and then transferred to multiple established vision and language tasks, such as visual question answering and visual common sense reasoning. Based on the MCAN and ReGAT models, we designed a deep co-attention visual object spatial relationship reasoning network. The spatial object reasoning feature constructed by the graph reasoning network is combined with the semantic feature of the visual object constructed by the deep co-attention module. Our model is better than the existing state-of-the-art visual question answering reasoning model according to the experimental results.

Table 4 compares the proposed SRRN model with the current state-of-the-art model on the VQA 2.0 dataset. Our model is superior to previous models with or without visual relational reasoning. The MCAN model is the champion model of the 2019 VQA Challenge. The SRRN model and the MCAN model have a better accuracy rate. Specifically, compared with the MCAN model, the accuracy of the four types has been improved (Yes/No increased by 0.15%, Number increased by 1.76%, Other increased by 0.08%, and All increased by 0.29% and 0.28% on *test-dev* and *test-std* respectively). It is worth noting that the "Number" type has increased by 0.98% and 0.6% compared with BAN-Counter [14] and ReGAT [20], respectively. It shows the effectiveness of our proposed spatial relationship reasoning based on graph neural networks.

Table 5 shows the comparison results between SRRN and the state-of-the-art model on the GQA dataset. The first block shows human performance, which can be considered the VQA task's upper bound. CNN+LSTM [23] uses a linear combination of image and question features to predict the answer. Other models use Faster-RCNN to extract image features. MAC is a milestone model on the CLEVR dataset, which decomposes a task into a series of continuous reasoning. SceneGraph [69] and LGCN [71] use graph neural networks to model the visual object region. The network completes the visual question answering task by jointly inferring visual objects' semantic and attribute relationships. DMFNet [70] uses a multi-graph inference and fusion layer to use pre-trained semantic relationships to embed inferences about complex spatial and semantic relationships between visual objects. According to Table 5, the SRRN

**Table 4. Performance comparison results on VQA 2.0 dataset.**

| Model | Test-dev | | | | Test-std |
| --- | --- | --- | --- | --- | --- |
| | Yes/No | Num | Other | All | All |
| Language only [22] | - | - | - | - | 44.26 |
| LSTM+CNN [22] | - | - | - | - | 54.22 |
| MCB reported in [22] | - | - | - | - | 62.27 |
| DCN [15] | 83.50 | 46.60 | 56.72 | 66.60 | 67.00 |
| Bottom-up [10] | 81.82 | 44.21 | 56.05 | 65.32 | 65.67 |
| Bottom-up+MFH [17] | 84.27 | 49.56 | 59.89 | 68.76 | - |
| MFH [68] | 85.31 | 49.56 | 59.89 | 68.76 | - |
| BAN [14] | 85.42 | 50.93 | 60.26 | 69.52 | - |
| BAN-Counter [14] | 85.42 | 54.04 | 60.52 | 70.04 | 70.35 |
| VRR [37] | 83.31 | 45.51 | 58.41 | 67.20 | 67.34 |
| DFAF [16] | 86.09 | 53.32 | 60.49 | 70.22 | 70.34 |
| MuRel [17] | 84.77 | 49.84 | 57.85 | 68.03 | 68.41 |
| ReGAT [20] | 86.08 | 54.42 | 60.33 | 70.27 | 70.58 |
| MCAN [17] | 86.82 | 53.26 | 60.72 | 70.63 | 70.90 |
| ViLBERT [31] | - | - | - | 70.55 | 70.92 |
| VisualBERT [34] | - | - | - | 70.80 | 71.00 |
| **SRRN-r-s-c(Ours)** | **86.97** | **55.02** | **60.80** | **70.92** | **71.18** |

model has better accuracy than the most advanced reasoning model. Compared with DMFNet [70], Accuracy, Open, Binary, and Consistency increased by 0.54%, 0.33%, 1.77%, and 0.34%, respectively, but Validity and Plausibility did not reach the best level. We guess that it is because it is effective to add a graph neural network to the original basic model to reason about the spatial position relationship. However, the validity and rationality of testing whether the answer is within the scope of the question is insufficient.

## 4.5 Visualization

In Fig 8, we describe the effect of our model through the visualization results on the VQA 2.0 and GQA datasets. The first visualization is to describe the model on the VQA 2.0 dataset. For example, the third image in the first column counts the number of elephants. The VRR and MCAN model counts are incorrect, while our model counts are correct. When counting elephants, if the visual object is not modeled by spatial reasoning, it is easy to superimpose the

**Table 5. Performance comparison results on GQA dataset.**

| Model | Accuracy | Open | Binary | Validity | Plausibility | Consistency |
| --- | --- | --- | --- | --- | --- | --- |
| Human [23] | 89.30 | 87.40 | 91.20 | 98.90 | 97.20 | 98.40 |
| CNN+LSTM [23] | 46.55 | 31.80 | 63.26 | 96.02 | 84.25 | 74.57 |
| Bottom-up [23] | 49.74 | 34.83 | 66.64 | 96.18 | 84.57 | 78.71 |
| MAC [23] | 54.06 | 38.91 | 71.23 | 96.16 | 84.48 | 81.59 |
| SceneGCN [69] | 54.56 | 40.63 | 70.33 | 95.90 | 84.23 | 83.49 |
| BAN [14] | 56.19 | 41.13 | 73.31 | 96.77 | **85.58** | 84.64 |
| DMFNet [70] | 57.05 | 41.86 | 73.98 | **97.62** | 84.87 | 86.98 |
| LGCN [71] | 56.10 | - | - | - | - | - |
| **SRRN-r-s-c(ours)** | **57.59** | **42.19** | **75.75** | 97.11 | 85.41 | **87.32** |

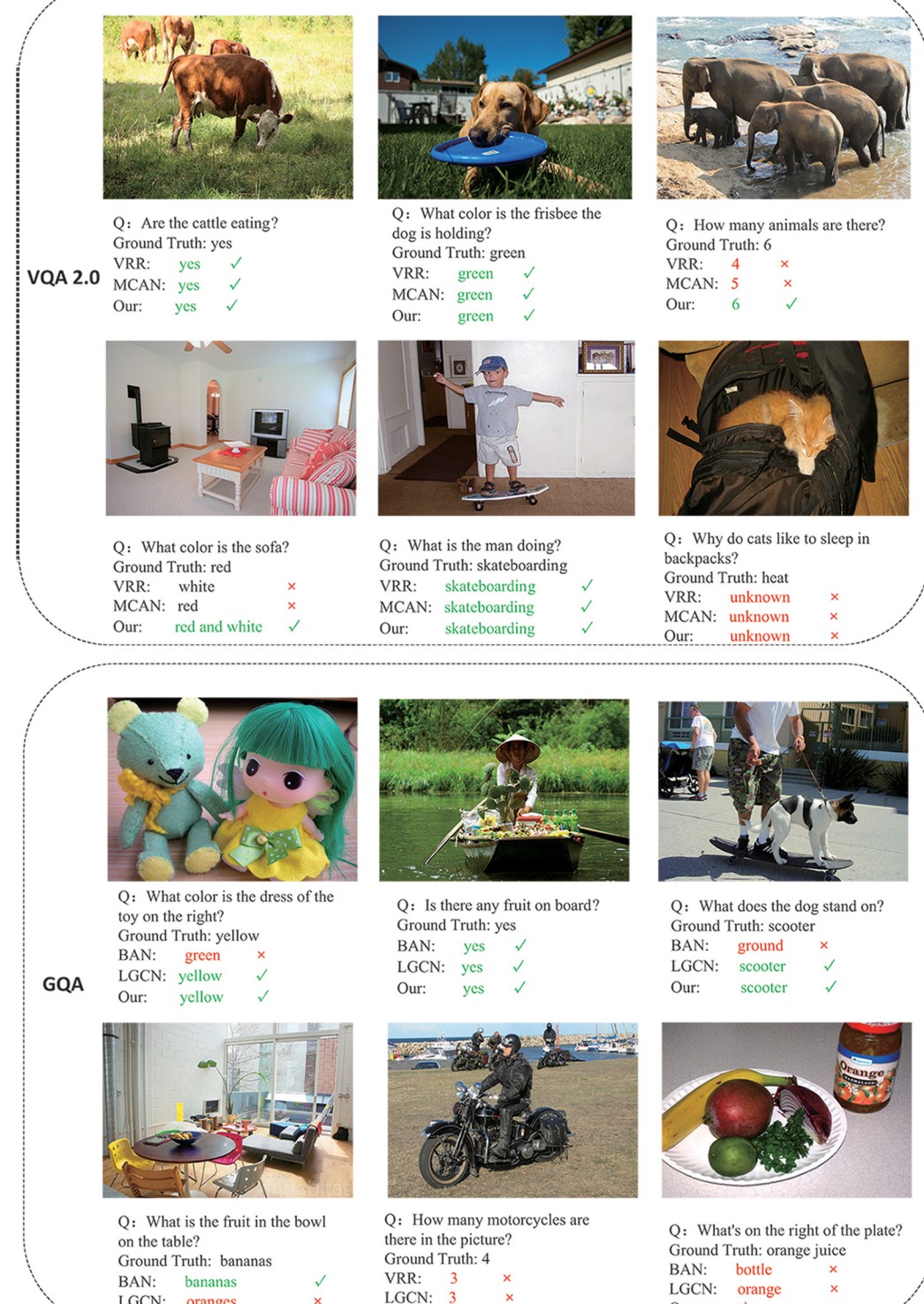

**Fig 8. Example of visualization on VQA 2.0 and GQA datasets.**

occluded elephant as a visual object, which will cause the model to count errors. Because our model integrates the object spatial relationship position reasoning module, it can effectively count in complex visual objects. It can also be seen from the experiment that our model has reached the current best level on the "Number" indicator. In the first image of the second row, our model answers correctly. Because the model combines semantic reasoning and spatial object-relational reasoning simultaneously, it better integrates object features, making the model more accurate in answering questions. However, in the last visualization example in the second line, the answers are wrong because of questions that require reasoning and external knowledge understanding. The model does not have an in-depth understanding of such questions and has external knowledge understanding, so it is difficult to answer them correctly.

In the GQA dataset visualization example, most of the questions require the model to understand and reason. For example, in the third picture in the first row, the model needs to understand the concepts of the two visual objects, the dog and the skateboard, and then infer the relationship between the dog and the skateboard to get the correct answer. Also, in the first picture in the second row, the LGCN model answered incorrectly, which is a spatial relationship reasoning question. The model must first find the positional relationship between the table and the space object of the plate and then understand the concept of "banana" in the plate through the positional relationship and semantic relationship to answer correctly. In the last example in the second row, our model answer is also wrong, which is a question that requires a deep understanding of both semantics and space. It is necessary to model the spatial position relationship of objects and understand the semantic features of the objects. It is also impossible to correctly understand some complex object semantic feature models. Therefore, it is difficult for the model to give a correct answer.

## 5. Conclusion

The VQA task requires the model to deeply understand visual objects' spatial position relationship features and semantic features. Existing methods generally focus on studying visual representations or interactive modeling of complex multimodalities. This paper investigates the importance of visual object spatial relational features for models to answer complex reasoning questions correctly. In addition, we also propose a sparse self-attention mechanism encoder, which can effectively capture contextual information while encoding the question while avoiding the introduction of irrelevant information in the modeling process. Finally, we utilize the compact self-attention (CSA) method to optimize the model, which effectively improves the accuracy and computational efficiency of the model based on the initial model. Experiments on our model on benchmark datasets VQA 2.0 and GQA demonstrate the effectiveness and interpretability of the SRRN model. Contribute to further research on spatial object relationship modeling for VQA tasks.

The accuracy of the SRRN model in each type of indicator in the VQA task has been improved. It can be seen from the experimental results that the addition of the spatial object relationship inference module has a significant improvement in the "Number" type indicator. The SRRN model solely focuses on the spatial position relationship of objects among many visual relationships. However, many interaction relationships between objects, such as behavioral relationships, represent action interactions. The follow-up work will explore more interaction relationships between objects and apply the relational reasoning method proposed in this paper to every visual relationship.

## Author Contributions

**Conceptualization:** Dezhi Han.

**Funding acquisition:** Dezhi Han, Gaofeng Luo.

**Investigation:** Zhongdai Wu.

**Methodology:** Xiang Shen.

**Resources:** Gaofeng Luo.

**Validation:** Xiang Shen, Chongqing Chen.

**Visualization:** Xiang Shen.

**Writing – original draft:** Xiang Shen.

**Writing – review & editing:** Xiang Shen, Dezhi Han.

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
