## [Decision Letter · Decision Letter 0]

26 Apr 2022

PONE-D-21-39541An effective spatial relational reasoning networks for visual question answeringPLOS ONE

Dear Dr. Han,

Thank you for submitting your manuscript to PLOS ONE. After careful consideration, we feel that it has merit but does not fully meet PLOS ONE’s publication criteria as it currently stands. Therefore, we invite you to submit a revised version of the manuscript that addresses the points raised during the review process.

ACADEMIC EDITOR: Please revised the paper based on the comments of the reviewers.

We look forward to receiving your revised manuscript.

Kind regards,

Sriparna Saha, PhD

Academic Editor

PLOS ONE

Journal Requirements:

“This research is supported by the National Natural Science Foundation of China under Grant 61873160, Grant 61672338, and the Natural Science Foundation of Shanghai under Grant 21ZR1426500.

This research is also supported by the Hunan Provincial Natural Science Foundation (Grant No. 2020JJ4557).”

“XS and DH are supported by the National Natural Science Foundation of China under Grant 61873160, Grant 61672338, and the Natural Science Foundation of Shanghai under Grant 21ZR1426500. https://isisn.nsfc.gov.cn/egrantindex/funcindex/prjsearch-list.

GL is supported by the Hunan Provincial Natural Science Foundation (Grant No. 2020JJ4557). http://kjt.hunan.gov.cn/kjt/zxgz/zkjj/xmgljcx/202006/t20200605_12266401.html”

“XS and DH are supported by the National Natural Science Foundation of China under Grant 61873160, Grant 61672338, and the Natural Science Foundation of Shanghai under Grant 21ZR1426500. https://isisn.nsfc.gov.cn/egrantindex/funcindex/prjsearch-list.

GL is supported by the Hunan Provincial Natural Science Foundation (Grant No. 2020JJ4557). http://kjt.hunan.gov.cn/kjt/zxgz/zkjj/xmgljcx/202006/t20200605_12266401.html”   

5. We note that Figure 1 in your submission contain copyrighted images. All PLOS content is published under the Creative Commons Attribution License (CC BY 4.0), which means that the manuscript, images, and Supporting Information files will be freely available online, and any third party is permitted to access, download, copy, distribute, and use these materials in any way, even commercially, with proper attribution. For more information, see our copyright guidelines: http://journals.plos.org/plosone/s/licenses-and-copyright.

  a.       You may seek permission from the original copyright holder of Figure(s) [#] to publish the content specifically under the CC BY 4.0 license.

5. We note that Figures 2 and 8 includes an image of a [patient / participant / in the study].

Additional Editor Comments (if provided):

The paper is based on the traditional approach to VQA with 71% accuracy. The recent approaches (pre-trained Transformer) have achieved performance of ~81% accuracy. The authors need to discuss those approaches and how (in terms of computational complexity, interoperability, low resource scenarios, etc. ) and when the proposed method will be useful.

Reviewers' comments:

Reviewer's Responses to Questions

**Comments to the Author**

1. Is the manuscript technically sound, and do the data support the conclusions?

Reviewer #1: Yes

Reviewer #2: Yes

2. Has the statistical analysis been performed appropriately and rigorously? 

Reviewer #1: Yes

Reviewer #2: Yes

3. Have the authors made all data underlying the findings in their manuscript fully available?

Reviewer #1: Yes

Reviewer #2: No

4. Is the manuscript presented in an intelligible fashion and written in standard English?

Reviewer #1: No

Reviewer #2: Yes

5. Review Comments to the Author

Reviewer #1: This work introduces the spatial and semantic reasoning network to learn the spatial position relationship and object attribute relationship of visual objects in VQA tasks. The authors show that their approach outperforms the attention and fusion-based existing methods on VQA 2.0 and GQA. However, there is no discussion on ongoing pre-trained Transformer based approaches. The pre-trained networks have pushed the boundaries of VQA performance. A detailed discussion on pre-trained networks and why one should use the proposed method (with less than ~10% overall accuracy compared to pre-trained approaches) is required for the reader to understand the usefulness of the approach.

Strengths:

The results show their approach improves the performance on the VQA 2.0 and GQA datasets compared to the non-pretrained network.

A detailed experiment and analysis are provided.

Weakness:

The paper proposed exciting ideas, but they can be presented in a much better way.

Related work on pre-trained network-based approaches and comparisons are missing.

Questions:

With n words in the question, Eq. (3) must obey the index. It can not be n+1. Please re-write the equation.

It is written (line 243-245) “Therefore, we use the question-adaptive attention mechanism to extract the semantic information of the question when designing the visual object spatial relationship graph reasoning network. “ However, the question-information is never incorporated to generate the reasoning feature v* of spatial object relation. Please clarify.

Function M in Eq. 10 is independent of the δ. Please re-write it. Authors should be consistent with the notation. The variable Q denotes multiple things (question and key) in the paper.

Eq. 11 and 12 are the same, and the description around them are hard to follow. If the matrix W has c row (ref. Line 309), then how come the vector t has t (again, please use another notation) row as written in “We connect the thresholds of each row to form a vector t = [t1, t2, . . . , tt ]”?

The function SAtt(;) is not defined. Please define and refer to the appropriate equation/section.

Can you provide the appropriate reference to your statement: “Since Q and K vary linearly from the same input, the weight matrices Wk and WQ are entangled in the gradient backpropagation, a basic redundancy in the conventional self-attention mechanism”

What do the authors mean by this statement:

“This paper proves that SRRN is indeed a virtual visual 628 object spatial relationship network.”

The contributions statement needs to be re-written. It is hard for the reader to understand.

What does this sentence refer to: “The ablation experiment analyzes the hypernatremia effect of the SSRN model.”

What is vg in Section 4.3?

In Table 1, the performance of SRRN-c is better than SRNN-s. Does the proposed sparse attention mechanism not better than the traditional one?

Each model variant can be better presented in the itemized format.

Typos/grammar:

“We input the word embedding sequence of size n × 300 to get the question, where n × 300 is the number of words contained in the question.”

I think n × 300 should be changed to n.

“Inattention will lead to the failure of relevant information extraction.”

Duplicate sentence line 348-351

Reviewer #2: Good

1. The research background is clearly articulated.

2. Distinct progress is shown at the paper.

To be improved

1. Results and Dataset are not available in the specified paths. (https://eval.ai/web/challenges/challengepage/

830/submission)

2. The architecture & training of Graph Neural Network can be elaborated.

Minor correction

1. Lines 348~351 are repeated.

6. PLOS authors have the option to publish the peer review history of their article (what does this mean?). If published, this will include your full peer review and any attached files.

Reviewer #1: No

Reviewer #2: No

---

## [Author Response · Author response to Decision Letter 0]

26 May 2022

Dear Editor,

 Thanks very much for taking your time to review this manuscript. We really appreciate all your generous comments and suggestion, which are very helpful to make this revision much clearer and more compelling. We have followed the comments and suggestions and revised the manuscript.

Thanks again for your patience and guidance!

Best regards,

Xiang Shen, Dezhi Han, Chongqing Chen, Gaofeng Luo, Zhongdai Wu

Response to Editor

Additional Editor Comments (if provided):

The paper is based on the traditional approach to VQA with 71% accuracy. The recent approaches (pre-trained Transformer) have achieved performance of ~81% accuracy. The authors need to discuss those approaches and how (in terms of computational complexity, interoperability, low resource scenarios, etc. ) and when the proposed method will be useful.

Response editor：

First of all, thank you very much for your comment. Pre-training is widely used in vision and language tasks, achieving surprising results in various tasks. Undoubtedly, pre-training can effectively facilitate alignment between different modalities, and reasonable fine-tuning in downstream tasks can achieve great results. It has reached 81% in the VQA task.

Response:

Although the pre-trained model achieves good results in a specific VQA task, many researchers also focus on developing end-to-end models. For public VQA benchmark datasets, there are VQA 1.0, VQA 2.0, GQA, VQA-E, CLEVR, etc. To verify the effectiveness of the proposed models, researchers usually use these benchmarks dataset for training. We employ end-to-end development because we can flexibly design and change the network structure without considering the effect of the model due to dataset bias or other factors. In addition, considering the hardware resources, if a pre-trained model is used to obtain a pre-trained model for a specific VQA task, the amount of datasets and better hardware resources are required for training. Currently, our hardware resources are insufficient to support pre-training, and pre-training also takes a long time. The final model is fine-tuned on the VQA task to achieve good results. If we directly utilize the existing pre-trained model and fine-tune it in a specific VQA task, it may also be due to the bias of the dataset and the differences between tasks. Without an excellent fine-tuning strategy, the final result may also be unsatisfactory. Since we changed the model structure of the network, we cannot rigorously prove the effectiveness of our model without using a specific dataset for training.

Of course, we are also considering using pre-trained models in future work. To ensure that our end-to-end model has a perfect effect, continuing to employ pre-training for fine-tuning will promote the development of the VQA task and get better results. In recent research work, there are also pre-trained models using similar network structures in this paper, such as ViLT-BERT, VisualBERT, and ViLBERT, among which the accuracies of ViBERT, VisualBERT, and ViLBERT are 70.34%, 70.80%, 70.55%, respectively. The SRRN model is better than these models.

Based on your comments, to allow readers to understand our research work better, we have added pre-trained models and explained them in the related work section and model comparison. Related pre-trained models have been discussed in the related work section (Section 2.1: lines134-159) and the model comparison section (Section 4.4: lines 611-615), with modifications marked in blue.

Response to Reviewer #1

Dear Reviewer#1:

Thank you for reviewing our paper entitled “An effective spatial relational reasoning networks for visual question answering. (ID:PONE-D-21-39541)” Also, we would like to thank you for your good comments and suggestions on it. We have studied each of your comments carefully and made corresponding revisions at your suggestion, which are highlighted in blue in this paper. All of your question were answered one-by-one.

Reviewer #1: This work introduces the spatial and semantic reasoning network to learn the spatial position relationship and object attribute relationship of visual objects in VQA tasks. The authors show that their approach outperforms the attention and fusion-based existing methods on VQA 2.0 and GQA. However, there is no discussion on ongoing pre-trained Transformer based approaches. The pre-trained networks have pushed the boundaries of VQA performance. A detailed discussion on pre-trained networks and why one should use the proposed method (with less than ~10% overall accuracy compared to pre-trained approaches) is required for the reader to understand the usefulness of the approach.

Response：

First of all thank you very much for your comment, as you said, “The pre-trained networks have pushed the boundaries of VQA performance.” In the past few years, the emergence of pre-training models has brought uni-modal fields such as computer vision and natural language processing to a new era. Substantial works have shown they are beneficial for downstream uni-modal tasks and avoid training a new model from scratch. Pre-training is an important technology in two aspects of computer vision and natural language, whose models can fine-tune different downstream tasks. Some representative methods, including ViLBERT, VLBERT, LXMERT, UNITER, OSCAR, VisualBERT, etc., use the bidirectional encoder representations from Transformers (BERT) structure and have achieved effective results in VQA tasks.

Why do we consider using an end-to-end network structure?

First, to demonstrate the SRRN model's effectiveness, we conveniently use the traditional end-to-end model to compare with the baseline model MCAN. The end-to-end model can be trained and tested on specific datasets (VQA 2.0 and GQA).

Second, for a specific VQA task. We train with a benchmark dataset for the VQA task, which can illustrate the rationality and effectiveness of the model more effectively. Most of the vision and language tasks that use pre-training today include multiple tasks, such as masked language modeling, masked object prediction (feature regression and label classification), cross-modality matching, and image question answering. Fine-tuning for downstream tasks requires meeting the demands of multiple tasks, and enabling parameter sharing is also a challenge.

Finally, although pre-trained models can achieve outstanding results on some tasks, there are certain disadvantages to using pre-trained models. For example, the pre-training model is large, the parameters are many, the flexibility of the model structure is poor, it is difficult to change the network structure, the calculation amount is large, and the application scenarios are limited.

Based on your comments, to allow readers to understand our research work better, we have added pre-trained models and explained them in the related work section and model comparison. Related pre-trained models have been discussed in the related work section (Section 2.1: lines 134-159) and the model comparison section (Section 4.4: lines 611-615), with modifications marked in blue.

Strengths:

The results show their approach improves the performance on the VQA 2.0 and GQA datasets compared to the non-pretrained network.

A detailed experiment and analysis are provided.

Weakness:

The paper proposed exciting ideas, but they can be presented in a much better way.

Related work on pre-trained network-based approaches and comparisons are missing.

Response：First of all, thank you very much for your affirmation of our research work and for pointing out the problems in our manuscript. In response to the issues in the manuscript, we carefully revise and improve it according to your comments, and it is your comments that make our research work and manuscripts be presented to readers in a better way.

Questions:

Q1. With n words in the question, Eq. (3) must obey the index. It can not be n+1. Please re-write the equation.

Response Q1:Thank you very much for pointing out the problems in our manuscript. Based on your comments, we re-checked the manuscript. To give readers a better understanding of the formulas in the manuscript, we use S to represent the maximum number of words so as not to conflict with n in Equation 3. Modifications in the manuscript are marked in blue as lines 264 to 269. Modified Equations and symbols are marked in yellow.

Q2. It is written (line 243-245) “Therefore, we use the question-adaptive attention mechanism to extract the semantic information of the question when designing the visual object spatial relationship graph reasoning network. “ However, the question-information is never incorporated to generate the reasoning feature v* of spatial object relation. Please clarify.

Response Q2: First of all, thank you very much for your comments. Section 3.2 mainly describes the visual object spatial relation inference module. The original meaning of what we want to express is: that the visual object spatial relationship combined with the visual semantic reasoning module can obtain image features with spatial relationships and object semantic attributes.The question information is not used in the spatial object reasoning module. Only in the semantic relation reasoning module is semantic information utilized to guide the attention to the vital information of the image. The sparse attention mechanism in the visual object semantic reasoning module can adaptively obtain critical visual semantic features according to the question information. In order not to cause misunderstanding and ambiguity to the readers, according to your suggestion, we remove the description of the information about the semantic reasoning of visual objects in this section3.2 (lines292-294).

Q3. Function M in Eq. 10 is independent of the δ. Please re-write it. Authors should be consistent with the notation. The variable Q denotes multiple things (question and key) in the paper.

Response Q3: First of all, thank you very much for your comment, based on your comments. To allow readers to understand the meaning of the equation(10) better, we have re-worked the equation in the manuscript. Specifically: Rewrite the formula in Eq. 10, delete δ. Redefine the vector-matrix A. Modifications in the manuscript are marked in yellow as lines 369 to 372.

According to your comments, Q appears several times in the manuscript because Q, K, and V are used in both the encoder and decoder. To allow readers to understand Q's meaning in each chapter clearly, we use different subscripts to indicate different meanings. In the revised manuscript, Q used in the encoder is changed to QE, and V and K are also represented in the same way. Q, K, and V also appear in Section 3.3.2. To illustrate the compact self-attention principle, we still use the original dot-product self-attention mechanism formulation method because the compact self-attention(CSA) mechanism also uses the encoder and decoder. Modifications in the manuscript are marked in yellow as Line352-350 and lines 380-387. In addition, the question feature vector is represented by Qq.

Q4. Eq. 11 and 12 are the same, and the description around them are hard to follow. If the matrix W has c row (ref. Line 309), then how come the vector t has t (again, please use another notation) row as written in “We connect the thresholds of each row to form a vector t = [t1, t2, . . . , tt ]”?

Response Q4: First of all, thank you very much for your comments. I am very sorry for the inconvenience caused to your review due to the repeated formulas caused by our carelessness in typesetting. Based on your comments. We have re-changed the repeated equation (12) and made the definition and interpretation of the equation.To make the formulas in the manuscript concise and easy to understand, we also rewrite the formulas that share the same symbols and re-denote the vector-matrix originally represented by t in the manuscript with A. And also modify fig4 in lines 369-370. 

Q5. The function SAtt(;) is not defined. Please define and refer to the appropriate equation/section. Can provide the appropriate reference to your statement: “Since Q and K vary linearly from the n yousame input, the weight matrices Wk and WQ are entangled in the gradient backpropagation, a basic redundancy in the conventional self-attention mechanism”.

Response Q5: Thank you very much for your comment. Based on your comment, we have redefined the formula SAtt(;) in line389. Besides, our proposed compact self-attention mechanism is also inspired by the literature [21], and the cited reference in line 398 of the manuscript explains the compact self-attention mechanism(CSA). In the VQA task, we have found this simple but very effective method through many experiments. So we use this method in this paper, which can also inspire readers to use the compact self-attention mechanism to improve the model accuracy in similar VQA tasks.

Q6. What do the authors mean by this statement: line 628

“This paper proves that SRRN is indeed a virtual visual object spatial relationship network.”

Response Q6: Thank you very much for pointing out that sentences in our manuscript are complicated for readers to understand, based on your prompt. We rechecked and revised, and rewritten the Conclusion section. Sections re-written in the manuscript's Conclusions are marked in blue.

Q7. The contributions statement needs to be re-written. It is hard for the reader to understand.What does this sentence refer to: “The ablation experiment analyzes the hypernatremia effect of the SSRN model.”

Response Q7: Thank you very much for your comments, based on your comments. For readers to better read and understand our research work, we re-written the contributions of this paper. Modifications in the manuscript are marked in blue as lines 104 to 118.

Q8. What is vg in Section 4.3?

Response Q8:Visual genome (vg) is a dataset, a knowledge base, an ongoing effort to connect structured image concepts to language. We have explained the VG dataset in lines 528 and 529 of the manuscript.To effectively verify the experiment, three modes can be used during the experiment：--SPLIT={'train', 'train+val', 'train+val+vg'} can combine the training datasets. The default training split is 'train+val+vg'. Setting --SPLIT='train' will trigger the evaluation script to run the validation score after every epoch automatically. The download address of the vg dataset : https://pan.baidu.com/s/1QCOtSxJGQA01DnhUg7FFtQ#list/path=%2F

Q9. In Table 1, the performance of SRRN-c is better than SRNN-s. Does the proposed sparse attention mechanism not better than the traditional one?

Response Q9: First of all, thank you very much for your comments. In Table 1, "SRRN-c" is the experimental effect when we only utilize the compact self-attention mechanism without employing the sparse self-attention encoder and the spatial position relationship reasoning module. From the experimental results, It can be seen that although the compact self-attention mechanism has a simple idea, it has a good effect on improving the model's performance. Our baseline model is MCAN, and it can be seen from Table 1 that the impact of using a sparse attention mechanism (SRRN-s) in the encoder is better than that of the MCAN model, indicating that our proposed sparse attention mechanism is effective.

Additionally, the main goal of our research is to explore the use of suitable parameters in sparse self-attention combined with visual object spatial relational reasoning modules. As shown in Table 1, when only the compact self-attention mechanism is used, the change in the "Number" indicator is not apparent, although the overall effect is good. However, our proposed visual object spatial relationship reasoning aims to study the spatial position relationship between objects and objects, so improving the "Number" indicator in the experimental results is needed to prove the method's effectiveness.Therefore, we combine the sparse attention mechanism and the visual object spatial relation inference module for research, and the experimental results also demonstrate the effectiveness of our two methods. Finally, the model can be further optimized if the compact self-attention mechanism is added. The Number indicator of "SRRN-r1-s-c" in Table 1 has reached 55.02%, which has surpassed some models that focus on counting (Number), such as ReGAT [20 ] and BAN-Counter [15] et al.

Q10. Each model variant can be better presented in the itemized format.

Response Q10：Thank you very much for your comments, based on your comments. We re-arrange and breakdown the data results for Table 1.

Q11. “We input the word embedding sequence of size n × 300 to get the question, where n × 300 is the number of words contained in the question.” I think n × 300 should be changed to n.

Response Q11：First of all, thank you very much for checking the manuscript carefully and carefully for us. I am sorry that there are errors in the manuscript due to our hand mistakes. According to your suggestion, for readers to understand the steps of our research very clearly. We have reworked the content, marked in yellow (line 264-269 marked blue).

Q11. “Inattention will lead to the failure of relevant information extraction.” Duplicate sentence line 348-351.

Response Q11: Thank you very much for pointing out the problems in our manuscript, based on your hints. We checked and rechecked the manuscript, and now repeated sentences have been removed, and ambiguous sentences have been rewritten. The specific revisions have been marked in blue (line342-343) the manuscript. Repeated sentences on lines 348-351 have been re-corrected.

Response to Reviewer #2

Reviewer #2: Good

1. The research background is clearly articulated.

2. Distinct progress is shown at the paper.

Dear Reviewer#2:

First of all, thank you very much for your affirmation and encouragement of our research work, and thank you for reviewing our paper entitled “An effective spatial relational reasoning networks for visual question answering. (ID:PONE-D-21-39541)”. We are very sorry for the inconvenience caused to your review due to some defects in our article. We re-checked the manuscript and ensured that all data and formulas were utterly correct. Again, thank you very much for your affirmation of our work.

Q1. Results and Dataset are not available in the specified paths. (https://eval.ai/web/challenges/challengepage/830/submission)

Response Q1: Thank you very much for your comments, based on your comments. To ensure the accuracy of our experimental data, we upload our experimental code and some test models to the Baidu network disk. Among them, the VQA 2.0 dataset evaluation website: https://eval.ai/auth/login Username: shenxiang Password: 123456 VQA 2.0 dataset website:：https://pan.baidu.com/s/1C7jIWgM3hFPv-YXJexItgw#list/path=%2F

The data underlying the results presented in the study are available from (include the name of the third party https://visualqa.org/vqa_v2_teaser.html).

Q2. The architecture & training of Graph Neural Network can be elaborated.

Response Q2: Thank you very much for your comment. In this paper, we use the principles of graph neural networks in our spatial object-relational reasoning network. Specifically, we use the different visual object targets in the image as different nodes as the graph neural network as input and the relationship between different objects as the edge as the graph neural network. It is explained in Section 3.2 of the manuscript how to calculate the relation weights between different objects and finally get visual features with spatial relation positions.

Q3. Lines 348~351 are repeated.

Response Q3: Thank you very much for your careful examination of our manuscript, according to your request. We have removed duplicate sentences.

---

## [Decision Letter · Decision Letter 1]

12 Sep 2022

PONE-D-21-39541R1An effective spatial relational reasoning networks for visual question answeringPLOS ONE

Dear Dr. Han,

Thank you for submitting your manuscript to PLOS ONE. After careful consideration, we feel that it has merit but does not fully meet PLOS ONE’s publication criteria as it currently stands. Therefore, we invite you to submit a revised version of the manuscript that addresses the points raised during the review process.

We look forward to receiving your revised manuscript.

Kind regards,

Sriparna Saha, PhD

Academic Editor

PLOS ONE

Journal Requirements:

Additional Editor Comments (if provided):

One of the reviewers has suggested some minor changes for the paper. The authors are requested to incorporate these changes in the revised version of the paper.

Reviewers' comments:

Reviewer's Responses to Questions

**Comments to the Author**

1. If the authors have adequately addressed your comments raised in a previous round of review and you feel that this manuscript is now acceptable for publication, you may indicate that here to bypass the “Comments to the Author” section, enter your conflict of interest statement in the “Confidential to Editor” section, and submit your "Accept" recommendation.

Reviewer #1: (No Response)

Reviewer #2: All comments have been addressed

2. Is the manuscript technically sound, and do the data support the conclusions?

Reviewer #1: Yes

Reviewer #2: Yes

3. Has the statistical analysis been performed appropriately and rigorously? 

Reviewer #1: Yes

Reviewer #2: N/A

4. Have the authors made all data underlying the findings in their manuscript fully available?

Reviewer #1: Yes

Reviewer #2: No

5. Is the manuscript presented in an intelligible fashion and written in standard English?

Reviewer #1: Yes

Reviewer #2: Yes

6. Review Comments to the Author

Reviewer #1: I read the author's response to the pre-trained models' question. The computational resource issue is a valid concern for academia. But again, many of the models do not take many resources (GPUs) and time only to fine-tune them on a particular dataset. Does it mean the author proposed model is less complex and has fewer parameters than the existing pre-trained models?

In this case, it will be good to compare parameters and time taken to train (fine-tune) the model (if computational resources allow) between the proposed model and pre-trained models such as ViLBERT, VisualBERT, etc.

The dataset bias issue with the pre-trained model also needs to be discussed in the main paper.

Line 158 "....many researchers are still based on the end-to-end training method..." spelling mistakes and appropriate citations are missing.

The remaining concerns have been addressed. However, proofreading is required, and the uses of notation and equations need to be checked carefully.

Reviewer #2: (No Response)

7. PLOS authors have the option to publish the peer review history of their article (what does this mean?). If published, this will include your full peer review and any attached files.

Reviewer #1: No

Reviewer #2: No

---

## [Author Response · Author response to Decision Letter 1]

29 Sep 2022

Manuscript ID: PONE-D-21-39541R1

Paper Title: An effective spatial relational reasoning networks for visual question answering

Dear Editor,

Thank you for your letter and for the reviewers’ comments concerning our manuscript entitled “An effective spatial relational reasoning networks for visual question answering” (PONE-D-21-39541R1). These comments are all valuable and very helpful for revising and improving our paper, as well as the important guiding significance to our research. We have carefully studied the comments point-by-point and revised the paper accordingly.

Thanks again for your patience and guidance!

Best regards,

Xiang Shen, Dezhi Han, Chongqing Chen, Gaofeng Luo, Zhongdai Wu

Journal Requirements Response:

Response: First of all, thank you very much for your comments. According to the journal requirements, we rechecked the references cited in this article. The references we cite are not retracted papers. In addition, according to the requirements of reviewer #1, we re-added the pre-training model related to this paper and compared the parameter quantity and accuracy of the SRRN model proposed in this paper. For the convenience of reviewing, the newly added references are marked in yellow.

Reviewer's Responses to Questions

Comments to the Author

1. If the authors have adequately addressed your comments raised in a previous round of review and you feel that this manuscript is now acceptable for publication, you may indicate that here to bypass the “Comments to the Author” section, enter your conflict of interest statement in the “Confidential to Editor” section, and submit your "Accept" recommendation.

Reviewer #1: (No Response)

Reviewer #2: All comments have been addressed

Response: Thanks again to the editors and reviewers for their comments and suggestions on our paper. We carefully thought about and responded to all the questions raised, and marked them with different colors in the manuscript for easy review.

2. Is the manuscript technically sound, and do the data support the conclusions?

Reviewer #1: Yes

Reviewer #2: Yes

Response: I am very grateful for your affirmation of our work.

3. Has the statistical analysis been performed appropriately and rigorously?

Reviewer #1: Yes

Reviewer #2: N/A

Response：Thank you very much for reviewing and commenting on our paper again, in order to allow readers to better understand the method we propose. Based on editor and reviewer comments, we further refined the article and marked it in the manuscript.

4. Have the authors made all data underlying the findings in their manuscript fully available?

Reviewer #1: Yes

Reviewer #2: No

Response：Thank you very much for your review and comments on our article again. In order to allow readers to better understand and reproduce our proposed method, and conduct in-depth research, The code will be available at https://github.com/shenxiang-vqa/SRRN.

5. Is the manuscript presented in an intelligible fashion and written in standard English?

Reviewer #1: Yes

Reviewer #2: Yes

Response：I am very grateful for your affirmation of our work.

6. Review Comments to the Author

Reviewer #1: I read the author's response to the pre-trained models' question. The computational resource issue is a valid concern for academia. But again, many of the models do not take many resources (GPUs) and time only to fine-tune them on a particular dataset. Does it mean the author proposed model is less complex and has fewer parameters than the existing pre-trained models?

In this case, it will be good to compare parameters and time taken to train (fine-tune) the model (if computational resources allow) between the proposed model and pre-trained models such as ViLBERT, VisualBERT, etc.

Response: First of all, thank you very much for reviewing our paper again and giving us suggestions. Based on your suggestions and comments, we can better improve the article. As you said, most of the existing multi-modal pre-training models only need to be fine-tuned on specific task datasets to get good results, and pre-training models are gradually becoming the mainstream method for multi-modal tasks.

Based on your suggestion, we investigate and study the task of pretraining models for visual question answering. Through investigation and research, it is found that pre-trained models have their advantages. However, some pre-trained models have more parameters than our proposed (SRRN) model, and they are not as good as end-to-end models on visual question answering tasks. As shown in Table 1, we compare the existing classical visual question answering pre-training model with the SRRN model, and the SRRN model has advantages over the pre-training model in terms of both the amount of parameters and the accuracy. Through experimental comparison, it is found that we use the end-to-end model not only in terms of the number of parameters and far lower than the pre-trained model, but only 1 TitanX GPUs are required to train the model. It shows that the parameters and complexity of our proposed model are better than some existing pre-trained models. (Tabular data refer to Table 1 in the “Multi-stage Pre-training over Simplified Multimodal Pre-training Models”.)

model parameter Test-dev Test-std

Unified VLP - 70.50 70.70

VilBERT 218.9M 70.55 70.92

VisualBERT 85.05M 70.80 71.00

VL-BERT 134.8M 71.16 -

DFAF-BERT 173.2M 70.59 70.81

MLI-BERT 120.0M 71.19 71.27

SRRN-r2-s 58.98M 70.73 -

SRRN-r1-s 58.19M 70.92 71.18

According to your comment, we added the model parameters and model accuracy in the ablation experiment part of the paper (Section 4.3.3) to compare with some existing VQA pre-training models. Although the proposed model adds the spatial object relation inference module, it does not increase the parameter quantity and complexity of the model, and good results can be obtained by training under the same experimental conditions. The parameters trained by the two models are shown in the figure below, where the left side represents the parameter quantity for training the "SRRN-r1-s" model, and the right side represents the parameter quantity for training the "SRRN-r2" model. Section 4.3.3 that we have added to the manuscript are marked in light blue.

The dataset bias issue with the pre-trained model also needs to be discussed in the main paper.

Response：First of all，thank you very much for your question, based on your suggestion. We discussed the issue of dataset bias for pretrained models in the paper. The pre-trained datasets come from various corpora, and the features learned from different corpora have generalization and generality, and can be better used for different multimodal tasks after fine-tuning. We use an end-to-end model for visual question answering tasks through specific datasets (such as VQA 2.0 and GQA), making it easier to capture and extract text and image features, increasing the robustness of the model. Re-added in Section 4.1, in lines 443 to 450 of the manuscript.

Line 158 "....many researchers are still based on the end-to-end training method..." spelling mistakes and appropriate citations are missing.

Response：First of all, I am very sorry for the spelling mistakes in the manuscript due to our carelessness, and thank you very much for pointing out our problems. Based on your suggestion, we carefully check and correct the errors in the manuscript. Re-edited content in lines 129 to 132 of the manuscript.

The remaining concerns have been addressed. However, proofreading is required, and the uses of notation and equations need to be checked carefully.

Reviewer #2: (No Response)

Response：Thank you again for your careful review and comments on our paper, based on your suggestions. We again carefully check the manuscript to ensure the correctness of the manuscript.

---

## [Decision Letter · Decision Letter 2]

2 Nov 2022

An effective spatial relational reasoning networks for visual question answering

PONE-D-21-39541R2

Dear Dr. Han,

We’re pleased to inform you that your manuscript has been judged scientifically suitable for publication and will be formally accepted for publication once it meets all outstanding technical requirements.

Kind regards,

Sriparna Saha, PhD

Academic Editor

PLOS ONE

Additional Editor Comments (optional):

Reviewers' comments:

Reviewer's Responses to Questions

**Comments to the Author**

1. If the authors have adequately addressed your comments raised in a previous round of review and you feel that this manuscript is now acceptable for publication, you may indicate that here to bypass the “Comments to the Author” section, enter your conflict of interest statement in the “Confidential to Editor” section, and submit your "Accept" recommendation.

Reviewer #1: All comments have been addressed

2. Is the manuscript technically sound, and do the data support the conclusions?

Reviewer #1: Yes

3. Has the statistical analysis been performed appropriately and rigorously? 

Reviewer #1: Yes

4. Have the authors made all data underlying the findings in their manuscript fully available?

Reviewer #1: Yes

5. Is the manuscript presented in an intelligible fashion and written in standard English?

Reviewer #1: Yes

6. Review Comments to the Author

Reviewer #1: All the comments have been addressed by the authors. However, the GitHub code is incomplete any files/packages are missing e.g. core package, run.py etc. Please make the code repository in a runnable state.

7. PLOS authors have the option to publish the peer review history of their article (what does this mean?). If published, this will include your full peer review and any attached files.

Reviewer #1: No

---

## [Editor Report · Acceptance letter]

16 Nov 2022

PONE-D-21-39541R2 

An effective spatial relational reasoning networks for visual question answering 

Dear Dr. Han:

I'm pleased to inform you that your manuscript has been deemed suitable for publication in PLOS ONE. Congratulations! Your manuscript is now with our production department. 

Kind regards, 

on behalf of

Dr. Sriparna Saha 

Academic Editor

PLOS ONE